# SCALING WITH COLLAPSE: EFFICIENT AND PREDICTABLE TRAINING OF LLM FAMILIES

**Shane Bergsma   Bin Claire Zhang   Nolan Dey   Shaheer Muhammad   Gurpreet Gosal**
**Joel Hestness**
Cerebras Systems
{shane.bergsma,joel}@cerebras.net

## ABSTRACT

Effective LLM training depends on predictable scaling of key quantities—such as final loss and optimal hyperparameters—with model and dataset size. Qiu et al. (2025) recently showed that this predictability can extend beyond scalars: whole training loss curves can *collapse* onto a universal trajectory after a simple normalization. What remains unclear is whether this phenomenon persists for LLM families trained under *practical scaling recipes*, where width, depth, learning rate, batch size, and weight decay are scaled jointly. We show that it does: loss curves collapse across scales precisely when optimization hyperparameters are set optimally for the given data budget, in accordance with recent empirical scaling laws. Collapse therefore emerges as a signature of compute-efficient training. We demonstrate two applications at scale: (1) deviation-from-collapse provides a sensitive, early diagnostic of training pathologies, and (2) predictability of collapsed curves enables early stopping in large-scale hyperparameter tuning. Finally, we train a competitive LLM family, *Celerity*, using these insights, establishing collapse as an effective tool for developing efficient LLMs.

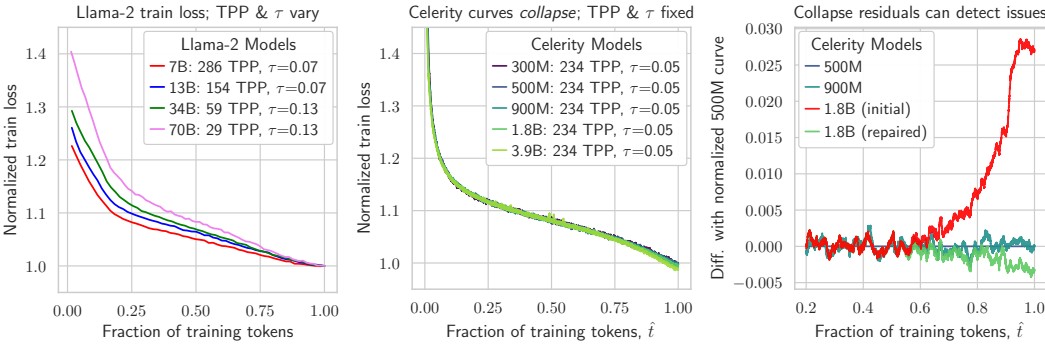

Figure 1: *Left*: Prior LLM families like Llama-2 train at varying tokens-per-parameter (TPP; $D/N$) and AdamW timescale $\tau$; training loss curves do *not* collapse. *Middle*: Fixing TPP and setting $\tau$ optimally for that TPP, Celerity loss curves *do* collapse. *Right*: Deviations from collapse allow precise identification (and earlier repair) of numerics issues in large-scale training runs.

## 1   INTRODUCTION

Scaling up pre-training has emerged as the primary route to improving LLM performance (Brown et al., 2020; Achiam et al., 2023). Yet once we reach frontier scales, opportunities for direct experimentation disappear (Xiao, 2024). How then can we train effectively at those scales—what size of model should we use, and how should we set hyperparameters? Encouragingly, recent work has revealed that several quantities are remarkably *predictable* as we scale deep learning. These include model performance as a function of model and dataset size (Hestness et al., 2017; Kaplan et al., 2020), as well as hyperparameters under maximal update parameterization ($\mu$P), which enables optimal base learning rates and initializations to approximately transfer across widths (Yang et al., 2021). In this paper we build on this trajectory of predictability: we show that, at LLM scale,

training loss curves (TLCs) from different model sizes *collapse* onto a single universal curve after a simple normalization—provided models are trained with a particular hyperparameter-scaling recipe.

Qiu et al. (2025) only recently demonstrated this striking regularity in TLCs, showing collapse when training with $\mu$P on small-scale autoregressive tasks. As their validation was limited to small models trained with vanilla Adam (Kingma & Ba, 2014), without weight decay, they explicitly call for tests at larger scales with *practical scaling ladders* that co-scale width, depth, batch size, and weight decay. Our work addresses this gap, showing that collapse persists in full-scale LLM families.

While modeling LLM loss is an active research topic (Sec. 6), the ability to predict TLCs has great *practical* value. For example, human judgment is now required to decide whether training has *recovered* from a loss spike—or whether rewinding/restarting is needed (Chowdhery et al., 2022; Zhang et al., 2022a). Other subjective signals, such as a gradual upward *trend* (Zhang et al., 2022b), can also trigger interventions. Yet criteria remain vague: Touvron et al. (2023b) report Llama-2 TLCs "did not show any sign of saturation," but how to recognize saturation is unclear. If TLCs collapse across sizes, practitioners can compare in-progress training to a universal reference, monitor residuals, and extrapolate final loss from partial trajectories. Teams already rely on TLCs in this way, often without a principled account of what governs TLC shape; for example, Falcon's final LR was chosen by simply continuing the run performing best after warmup (Almazrouei et al., 2023).

In this paper we show that the essential condition for collapse under $\mu$P is that the LR schedule, tokens-per-parameter ratio (TPP), and AdamW timescale $\tau$ (Wang & Aitchison, 2024) are held fixed across model sizes. This reflects a deeper regularity: prior work showed that optimal $\tau$ depends only on TPP (Bergsma et al., 2025a). Thus, scaling across fixed TPP with $\tau$ chosen optimally guarantees collapse, and collapse emerges as a robust marker of compute-efficient and stable pre-training. When $\tau$ is mis-scaled—as in the Llama-2 family (Fig. 1, *left*)—normalized curves fail to align.

We introduce *Celerity* as the first LLM family trained with both optimal $\tau$ scaling and demonstrable TLC collapse (Fig. 1, *middle*). Effective parameterization, including tuning and transferring $\tau$, helped Celerity land on the compute-efficiency frontier for open models of its scale (Fig. 2). Meanwhile, deviations from collapse provided a sensitive diagnostic of training issues: in our 1.8B run, a numerical instability became evident from collapse residuals (Fig. 1, *right*) well before the raw TLC showed an upward trend (Fig. 6, *right*). Celerity exemplifies *scaling with collapse*: efficient, predictable training, across scales and throughout the run.

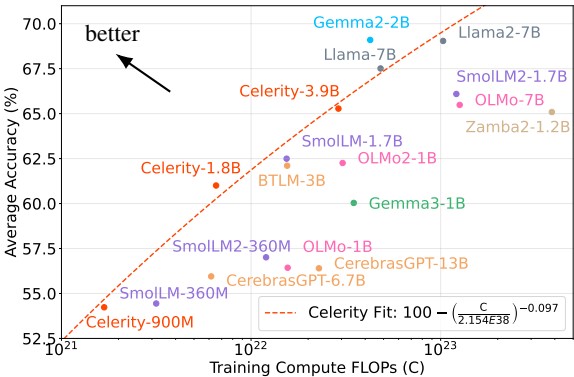

Figure 2: Celerity is at the compute-efficiency frontier. (Average accuracy on tasks *arc-c, arc-e, boolq, hellaswag, piqa, siqa, winogrande*; see Table 10.)

In summary, our main contributions are:

- Identifying the key factors influencing loss curve shape under $\mu$P: the LR schedule, the TPP ratio, and the AdamW timescale $\tau$, and explaining shape dependence on these quantities (Sec. 3).

- Demonstrating that when $\tau$ is set optimally for a given TPP, TLCs *collapse* across model scales, providing a signature of compute-efficient training (Sec. 3).

- Introducing the *Celerity* family, the first large-scale LLMs trained in a collapse regime (Sec. 4).

- Proposing a simple functional form for normalized TLCs, and showing that fitting this form on small-scale training runs enables early stopping in large-scale hyperparameter tuning (Sec. 5).

## 2 BACKGROUND

Training loss curves (TLCs) for different model sizes typically differ in scale, duration and final loss. Yet after a simple normalization, they can align closely, or *collapse*, onto a common trajectory. Collapse emerges only when three controls are matched across scales. The tokens-per-parameter

ratio (TPP) determines how much data each parameter sees and thus affects the normalized pace of improvement; the AdamW timescale $\tau$ governs how long the optimizer "remembers" past gradients, shaping the bias–variance trade-off over training; and the learning-rate schedule phases early bias reduction against late variance suppression. When TPP and $\tau$ are chosen consistently across model sizes and the LR schedule is fixed, the resulting normalized TLCs become approximately scale-invariant. We now formalize these quantities; Table 1 summarizes key symbols.

**TPP.** The TPP ratio is equal to number of training tokens $D$ divided by the model size $N$. This simple quantity plays a surprisingly profound role in compute-efficient LLM training and TLC shape. Hoffmann et al. (2022) investigated, for a given compute budget $C$, how to allocate $D$ and $N$ in order to minimize loss. They found optimal $D$ and $N$ scale roughly equally as $C$ increases, with the optimal $D/N$ ratio relatively constant at around 20 TPP (Appendix C.1). Replication studies have found similar results (Besiroglu et al., 2024; Porian et al., 2024), and 20 TPP has emerged as a rule-of-thumb for compute-optimal training (Dey et al., 2023a; Zhang et al., 2024b).

**$\mu$P.** $\mu$P (Yang & Hu, 2020) and related parameterizations for depth (Bordelon et al., 2023; Yang et al., 2023; Dey et al., 2025) seek to achieve consistent, stable training dynamics as networks scale up. Moreover, with $\mu$P, base hyperparameters can be tuned on a small *proxy* model and then transferred to larger scales. Given the width of the proxy model, $d_p$, and target, $d_t$, $\mu$P prescribes scaling factors to apply to the base LR, initial weight variance, and other base HPs.

$\mu$P is increasingly used in LLM training (Dey et al., 2023a;b; Sengupta et al., 2023; Shen et al., 2024; Hu et al., 2024). Moreover, recent work has shown that, when using $\mu$P, other important aspects of training may *decouple* from model size, including optimal batch size (scaling primarily in the total number of tokens (Zhang et al., 2024b; Bergsma et al., 2025a)), and optimal AdamW timescale/weight decay (scaling primarily in TPP (Bergsma et al., 2025a)).

**Supercollapse.** Using $\mu$P, Qiu et al. (2025) observed that TLCs for different model sizes, despite varying widely over compute and absolute loss, appear to follow a consistent shape. This motivated them to affinely rescale the curves to the normalized loss $\ell$ given by:

$$\ell(\hat{t}, N, \omega) = (L(\hat{t} \cdot T^{\star}(N), N, \omega) - \hat{L})/(L(T^{\star}(N), N, \omega) - \hat{L}) \tag{1}$$

where $\omega$ is the random seed, $\hat{t}$ is the fraction of training completed (what Qiu et al. (2025) refer to as *normalized compute*), $N$ is the number of model parameters, and $T^{\star}(N)$ is the corresponding compute-optimal number of training steps, estimated from a power law fit. $\hat{L}$ is an offset, which they subsequently set to the estimated irreducible loss of their power law.

Training compute-optimally under $\mu$P (on small-scale autoregressive tasks, e.g., predicting chess moves), Qiu et al. (2025) showed TLCs collapse under this normalization—indeed, they *super*collapse, meaning they differ by less than the noise from inter-run variation. They further show that collapse arises naturally in constant-learning-rate models where loss obeys typical neural power laws, while extending the theory to arbitrary LR schedules via a theoretical model of quadratic loss.

**The AdamW EMA and its timescale.** AdamW updates at step $t$ can be expressed in terms of learning rate $\eta$ and weight decay $\lambda$ as: $\theta_t = (1 - \eta\lambda)\theta_{t-1} - \eta\frac{\hat{m}_t}{\sqrt{\hat{v}_t}+\epsilon}$, where $\hat{m}_t$ and $\hat{v}_t$ are bias-corrected EMAs of gradients and squared gradients, respectively (Kingma & Ba, 2014). Wang & Aitchison (2024) observed that AdamW parameters $\theta_t$ can also be viewed as an EMA—of weight *updates*. That is, the standard EMA form $y_t = (1 - \alpha)y_{t-1} + \alpha x_t$ matches AdamW when $y_t = \theta_t$, $\alpha = \eta\lambda$, and $x_t = -\frac{1}{\lambda}\frac{\hat{m}_t}{\sqrt{\hat{v}_t}+\epsilon}$. The *timescale* $\tau_{\text{iter}} = 1/\alpha = 1/\eta\lambda$ represents the approximate number of iterations over which updates are averaged. When expressed in epochs as $\tau_{\text{epoch}} = \tau_{\text{iter}}/M$, where $M$ is the number of iterations per epoch, Wang & Aitchison (2024) found the optimal $\tau_{\text{epoch}}$ (swept by varying $\lambda$) *remains stable* under model and dataset scaling on image tasks.

Since LLM pre-training typically uses a single epoch, we follow Bergsma et al. (2025a) in defining a normalized timescale $\tau = \tau_{\text{iter}}/T$, where $T$ is the total number of optimization steps. As $T = D/B$ (total tokens/batch size):

$$\tau = 1/(\eta\lambda T) = B/(\eta\lambda D). \tag{2}$$

In contrast with the results in Wang & Aitchison (2024), Bergsma et al. (2025a) did *not* find optimal $\tau$ to remain stable in LLM training, but instead to decrease as a (scale-invariant) power law in TPP.

Table 1: Core quantities used throughout the paper.

| Symbol | Meaning |
| --- | --- |
| $N$ | Number of model parameters |
| $D$ | Total number of training tokens |
| $B$ | Batch size (tokens per optimization step) |
| $T = D/B$ | Total number of optimization steps |
| $\hat{t} = t/T$ | Fraction of training completed |
| TPP $= D/N$ | Tokens-per-parameter ratio (TPP) |
| $L(t)$ | Training loss at step $t$ |
| $\ell(\hat{t})$ | Normalized training loss curve (TLC) |
| $\eta$ | Learning rate |
| $\lambda$ | Weight decay coefficient |
| $\tau_{\text{iter}} = 1/(\eta\lambda)$ | AdamW timescale (in steps) |
| $\tau = \tau_{\text{iter}}/T = 1/(\eta\lambda T) = B/(\eta\lambda D)$ | Normalized AdamW timescale |

## 3 WHAT FACTORS MODULATE TRAINING CURVE SHAPE?

**Experimental setup.** We use a GPT2-like LLM (Radford et al., 2019), with ALiBi embeddings (Press et al., 2022) and SwiGLU (Shazeer, 2020). We train on SlimPajama (Soboleva et al., 2023). Models are trained with AdamW and $\mu$P. We use a linear decay-to-zero LR schedule, context length of 2048, and the GPT2 vocabulary. Full architecture and other details are in Appendix B.1.

We plot $\ell$ vs. *training fraction* $\hat{t} = t/T = tB/D$, with step count $t$, total steps $T$, batch size $B$, and dataset size $D$. To reduce noise in small-$B$ settings, we *post hoc* aggregate losses $\ell(\hat{t})$ using a moving-average filter over a window of 100 steps, smoothing curves without altering the underlying trajectory. We also consistently found simply *dividing by the final training loss* (i.e., $\hat{L} = 0$ in Eq. (1)) resulted in optimal alignment across scales, so use this for all curves.

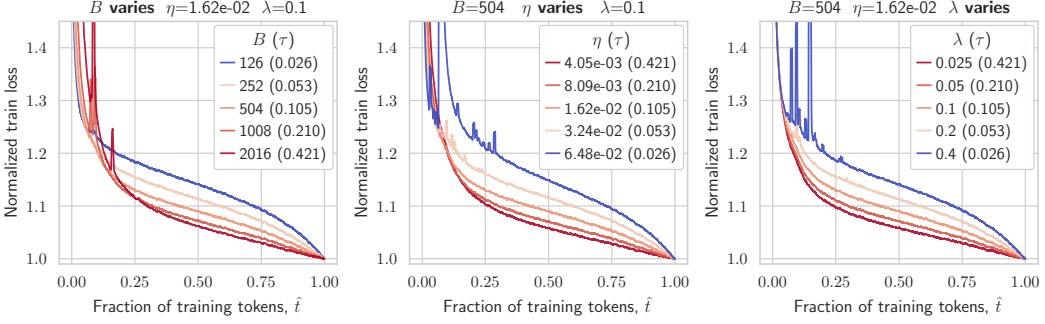

Figure 3: **AdamW timescale $\tau$ modulates TLC shape (610M, 80TPP)**: Sweeping $\eta$ (*left*), $\lambda$ (*middle*), or $B$ (*right*) produces matching variations in normalized TLCs when $\tau$ varies identically.

**Finding: $\tau$ modulates TLC shape.** Fig. 3 shows normalized TLCs for 610M models trained to 80 TPP, sweeping either learning rate $\eta$, weight decay $\lambda$, or batch size $B$ in each subplot. Across hyperparameters, TLCs with matching $\tau$ exhibit very similar shapes, reflecting consistent timescale control. Similar patterns hold across other scales and dataset sizes. Generally, as $\tau$ increases, TLCs drop more early and less later. This is also a function of the LR schedule: when we switch to using a *Constant* LR, there is no final drop, lower-$\tau$ TLCs are lower throughout (appendix Fig. 10).

**Finding: TPP modulates TLC shape.** We now fix $\tau$ and test increasing *TPP*, finding TLCs drop earlier and flatten for longer (Fig. 4, *left*, *middle*; see also Llama-2 for $\tau = 0.07$ and $\tau = 0.13$ in Fig. 1, *left*). Intuitively, relative to length of training, higher TPP drops loss more at the beginning and then obtains diminishing returns later on. In Fig. 4, *right*, TLC shape is quite similar across model scales at the same TPP (when $\tau$ is roughly equal), showing TPP's shaping effect is *scale-invariant* (scaling from 111M to 3.3B at fixed TPP represents a $1000\times$ increase in training FLOPs).

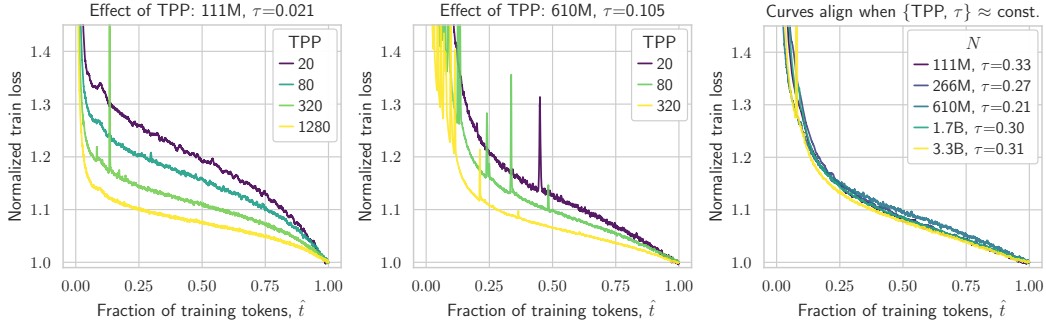

Figure 4: **TPP modulates TLC shape.** Fixing $\tau$ for 111M (*left*) & 610M (*middle*) while *increasing* TPP, curves shift *down*. When $\tau \approx$ const. and TPP also fixed (at 20), curves roughly collapse (*right*).

**Using bias and variance to explain the effect of $\tau$.** Classical analyses of stochastic optimization (e.g., Bottou et al. (2018); D'Angelo et al. (2024)) decompose loss into two components: a *bias* term, reflecting dependence on initial weights, and a *variance* term, reflecting noise from stochastic gradients. Early in training bias dominates; later, variance determines the attainable loss floor.

In our setting, the AdamW timescale $\tau$ controls this trade-off through the EMA over weight updates (Sec. 2). A smaller $\tau$ corresponds to a shorter memory: updates depend mainly on recent gradients, yielding rapid bias reduction but a higher variance floor. A larger $\tau$ averages over more past gradients, reducing variance more effectively but slowing early progress. Empirically, TLCs reflect this pacing, with smaller $\tau$ producing faster early descent (e.g., left panel of appendix Fig. 10).

This intuition is formalized in Appendix B.3 via a noisy quadratic model:

$$\mathbb{E}[L(\hat{t})] = \frac{h\,\sigma_x^2}{4\,\tau}\Big(1 - e^{-2\hat{t}/\tau}\Big) \;+\; \frac{h}{2}\,e^{-2\hat{t}/\tau}\,\mathbb{E}[\theta(0)^2], \tag{3}$$

where the first term approaches a variance floor $\propto 1/\tau$ and the second is an exponentially decaying bias term. Smaller $\tau$ thus yields faster initial decay but a higher floor, while larger $\tau$ yields slower initial decay but a lower floor, matching the observed *fast-then-flatten* behavior under constant LR. With LR decay, $\eta_t\lambda$ decreases and the instantaneous timescale $\tau_t = 1/(\eta_t\lambda T)$ *increases*, enhancing late-stage variance suppression and steepening final drop (e.g., Fig. 10: more LR decay, more drop).

*Scale invariance.* After normalizing by the final loss, the curvature factor $h$ cancels (Appendix B.3). Provided residual bias at end-of-training is negligible relative to the variance floor, the normalized TLC depends only on $\tau$ and $\hat{t}$. Thus, at matched $\tau$, normalized TLCs collapse across scales.

**Explaining effect of TPP.** TPP affects TLCs via power laws. With a *constant* LR, every step is the endpoint of a shorter run, trained to $\hat{t} \cdot$ TPP. Qiu et al. (2025) note $L(\hat{t})$ therefore *follows the same power law as final loss of a run fully trained to that effective budget*. Normalizing by the projected loss at *total* TPP removes dependence on model and dataset size, so $\ell(\hat{t})$ depends only on $\hat{t}$ and total TPP (Appendix B.2). Higher-TPP curves analytically decay faster and level off sooner. LR schedules deform the curves, but deformation is also scale invariant given consistent curvature of the loss landscape across model sizes under $\mu$P (Noci et al., 2024).

> ***Key takeaway 1**: TLC shape is governed by three scale-invariant controls: (1) AdamW timescale $\tau$ (bias–variance trade-off), (2) tokens-per-parameter ratio (TPP), which sets the relative pace of improvement, and (3) learning-rate schedule, which phases early and late loss reduction. When matched across model sizes, normalized TLCs follow the same trajectory—that is, they collapse.*

## 4 CELERITY: A COMPUTE-EFFICIENT MODEL FAMILY WITH COLLAPSE

We have established that collapse arises when $\tau$ and TPP are held fixed across model sizes. Meanwhile, prior work has shown optimal $\tau$ to depend only on TPP (Bergsma et al., 2025a). Here, we introduce a model family, *Celerity*, trained at fixed TPP and with $\tau$ chosen optimally for that TPP, i.e., a regime where collapse emerges naturally as a consequence of good training.

**Compute vs. parameter efficiency.** A key question for Celerity is which TPP to use: $\approx 20$ is compute-optimal (Sec. 2), while higher TPP means greater *parameter efficiency* (fewer parameters to obtain same loss). For small, inference-ready models, parameter efficiency is paramount, but such models are usually distilled (Tunstall et al., 2023; Wang et al., 2025) rather than pre-trained at high-TPP. *Our main interest is developing pre-training strategies for very large models.* As models scale, the relative importance of compute-efficiency increases—indeed, public families often have declining TPP as size increases (Touvron et al., 2023a; Biderman et al., 2023).

Yet even for the largest models, parameter efficiency remains valuable, e.g., when generating distillation logits or synthetic data. To choose Celerity's TPP, we analyze this trade-off: Appendix C.1 derives an expression for the extra compute required to compress a model to a fraction of the *size when training compute optimally*—while maintaining the same loss. This expression leverages power law fits from prior work. Fig. 5 plots the trade-off, where a TPP ratio of 234 is estimated to achieve a 62% reduction in parameters with only a 67% increase in total FLOPs (relative to 20 TPP). This is a responsible balance point, near what has been called the *critical model size*—the point where further increasing compute obtains massively-diminishing returns in parameter efficiency (De Vries, 2023) (e.g., doubling *our* FLOPs, to $3.34\times$ compute-optimal, reduces $N$ by only a further $\approx 11\%$).

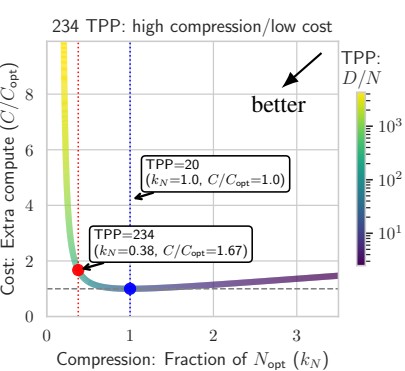

Figure 5: Expected iso-loss compute vs. compress trade-off as TPP varies.

Even if the ultimate goal is a "herd" of models at varying TPP, such as Llama-2 in Fig. 1, *left*, there are advantages to training different "bands" within the herd, e.g., 7B, 13B, 34B, 70B *all* at 29 TPP:

- *Tuning*: you can fine-tune $\tau$ at a smaller scale and zero-shot transfer to larger models.
- *Diagnostics*: Because TLCs collapse, deviations provide an early warning of training issues.
- *Cost*: Fixed-TPP bands are cheap (e.g., $10\times$ lower $N \to 10\times$ smaller $D \to 100\times$ less compute).

**Philosophy.** Celerity aims to advance general LLM capabilities using public pre-training corpora and fully-open, consistent methods—rather than targeting specific benchmarks. In contrast, the majority of LLMs now *anneal on training subsets of downstream benchmarks* (Dubey et al., 2024; Achiam et al., 2023), or inject special high-quality math (OLMo et al., 2024), code (Zhang et al., 2024a), or instruction (Hu et al., 2024) data during a late-stage *mid-training* process. Since these practices make evaluation problematic (Dominguez-Olmedo et al., 2024), Celerity can serve as a comparison for models trained without (or prior to) applying such techniques.

**Experimental details.** Celerity pre-trains in bands of 20, 80, and 234 TPP, each spanning 300M–3.9B models (Table 2); see Appendix C.2 for further details. Key enablers of Celerity's reliable, efficient training include:

- *Data*: emphasizing (open) educational, math, and coding data throughout training (appendix Table 6); this outperformed training on the general SlimPajama dataset (Table 7).

- *Parameterization*: Using CompleteP, which enables hyperparameter transfer over width *and* depth, was more efficient/reliable than $\mu$P (Fig. 15).

- *Optimization*: LR, $\tau$, batch size tuned small, transferred via scaling rules.

Table 2: Architecture of Celerity

| Celerity: | 300M | 500M | 900M | 1.8B | 3.9B |
|---|---|---|---|---|---|
| Hidden Dim | 640 | 896 | 1152 | 1536 | 2048 |
| Num Heads | 10 | 14 | 9 | 12 | 16 |
| Head Size | 64 | 64 | 128 | 128 | 128 |
| Layers | 13 | 17 | 23 | 30 | 40 |
| Batch Size | 176 | 240 | 336 | 464 | 672 |
| Vocabulary | Llama-3 (size 128256) | | | | |
| Embeddings | ALiBi, Untied | | | | |
| Seq Length | 8192 | | | | |
| Non-linearity | Squared ReLU | | | | |
| FFN Mult | $8\times$ | | | | |
| Norm Type | Pre-Layer Normalization, $\epsilon = 10^{-5}$ | | | | |
| LR Schedule | Peak: 0.15, linear decay-to-zero | | | | |
| LR warmup | min(10% of total tokens, 375M tokens) | | | | |

**Evaluation results.** Appendix Table 10 provides full downstream evaluation results for Celerity and other public models tested on seven common downstream tasks. Fig. 2 shows that Celerity models form the accuracy/compute Pareto frontier up to our largest training budget. Against BTLM (Dey et al., 2023b)—trained before task-specific data annealing became standard—Celerity achieves comparable accuracy with 75% fewer training FLOPs. Extrapolation via a fitted power law in compute (dashed line in plot) suggests smooth scaling and continued competitiveness. For comparison with distilled models, we count only *student* FLOPs in Fig. 2. Including *teacher* FLOPs (forward passes), or the cost of teacher training, strengthens Celerity further (appendix Fig. 16).

In terms of parameter efficiency, Celerity is weaker than high-TPP families (Figs. 19 and 20), meaning such models save FLOPs at inference. However, beyond the importance of studying compute efficiency for hyper-scale training, there is strong motivation to train and study compute-efficient smaller models: growing evidence suggests some models may be counter-productively (even *catastrophically*) overtrained, making them harder to fine-tune (Springer et al., 2025) and quantize (Kumar et al., 2024). Compute-efficient alternatives therefore serve both as a principled baseline for understanding scaling and as a practical fallback when high-TPP models prove brittle.

**Collapse results.** In Sec. 3, we normalized training loss curves by dividing by the final loss value, $L(T)$ (Eq. (1)). To use collapse as a diagnostic *during* training, we need a way to normalize when $L(T)$ is still unknown. We explored two strategies and use early-align in our experiments:

- *Estimate*: extrapolate $L(T)$ from a power law fit at lower scales.

- *Early-align*: choose $L(T)$ so $\ell(t)$ best aligns with the smallest-scale curve over 25-50% portion.

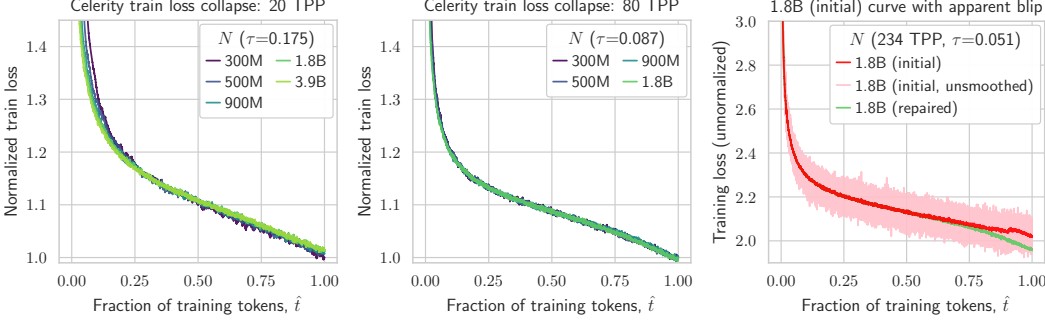

Figure 6: **Collapse in Celerity models.** Celerity 20 TPP (*left*) and 80 TPP (*middle*) models exhibit collapse. *Right*: smoothing helps detect blip in loss near the end of initial 1.8B run (red curve), but divergence can be detected much earlier using collapse residuals (Fig. 1, *right*).

Fig. 6 shows normalized curves. Collapse is tight at 80 TPP (*middle*). At 20 TPP (*left*), we see small early deviations, which we attribute to differing LR warmup proportions (Table 2). At 234 TPP, divergences appear late in training for larger models (Fig. 1, *middle*). Investigating, we find loss improves disproportionately on training data, while held-out data remains aligned with projections.

**Collapse for monitoring.** Fig. 6 (*right*) shows the unnormalized TLC for our original 1.8B, 234 TPP run. Smoothing helps reveal a sudden rise in training loss, but only after 90% of training. Without a collapse reference, it would be impossible to see that problems began much earlier. By comparing against the 500M TLC reference (Fig. 1, *right*), we pinpoint divergence starting near 60%. Knowing this timing was crucial: we did not waste effort investigating late-stage data redundancy, and instead realized the problem coincided with a job restart under a new compute allocation.

The collapse reference was also essential for debugging: by running ablations with different batch sizes and measuring divergence from the reference, we confirmed the anomaly arose from a numerical issue in a loss kernel triggered only at specific microbatch sizes. After fixing the kernel and restarting from before the divergence, training tracked the reference TLC closely (Fig. 1).

> ***Key takeaway 2****: Celerity trains models in fixed-TPP bands at the optimal $\tau$ for each band. This produces collapse across scales, enabling hyperparameter transfer, early detection of training issues, and reliable cross-scale comparisons. The 234-TPP band lies on the compute–accuracy frontier while using $\approx62\%$ fewer parameters than compute-optimal training at equal loss.*

## 5 COLLAPSE ENABLES EARLY STOPPING IN HYPERPARAMETER TUNING

Training to completion is expensive. If normalized TLCs behave predictably, can we stop earlier and still recover the final loss? We show collapse enables principled early stopping in tuning, and introduce a predictive model—fit at small scales, and re-used to extrapolate large-scale TLCs.

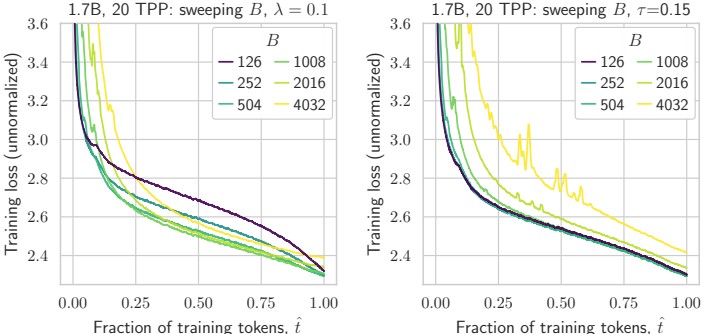

Figure 7: **Predictability of batch size sweeps.** *Left*: When fixing weight decay $\lambda$ in batch size sweeps (standard practice), normalized TLCs *cross*, making final loss hard to predict from partial results. *Right*: Fixing $\tau$ instead (by adjusting $\lambda$), TLCs maintain ordering, enabling early stopping.

**Role of $\tau$ in tuning.** Recent work tunes learning rate $\eta$ and batch size $B$ at smaller scales and extrapolates via power laws (Hu et al., 2024; Bi et al., 2024; Porian et al., 2024). These studies typically fix weight decay $\lambda$, which unintentionally varies $\tau$—and hence TLC shape. As Fig. 7 (*left*) shows, when $\tau$ varies, mid-training loss is a poor predictor of final outcomes during a batch size sweep. In contrast, when $\tau$ is fixed during tuning (by adjusting $\lambda$), the ordering of $B$-specific curves is preserved throughout training (*right*); runs can be stopped early (e.g., at 25%) while still reliably identifying the best batch size.

There are, however, cases where $\tau$ must vary. For example, Bergsma et al. (2025a) found optimal $\tau$ was no longer constant once $B > B_{\mathrm{crit}}$, potentially requiring retuning of $\lambda$.

**Exploiting collapse for early stopping.** Suppose we wish to find the optimal setting of a hyperparameter (HP) in large-scale training. Naively, we could sweep across settings of the HP and train a large-scale model to completion at each setting; the lowest final loss would identify the best choice. Alternatively, rather than training to completion, we propose a procedure to infer the final loss values from partial training runs. We do this by exploiting the collapse phenomenon as follows:

1. For each large-scale setting in the sweep, first identify the corresponding *TLC controls*, i.e, the LR schedule, $\tau$, and TPP.
2. Train a smaller (e.g., 100M-parameter) model for each unique combination of TLC controls.
3. Normalize these small-run loss curves to obtain the *normalized universal TLCs* ($\ell(\hat{t})$) corresponding to each set of controls.
4. Perform *partial* training runs (e.g., to 30% of tokens) at each HP setting.
5. For each partial large-scale loss curve, use its corresponding universal TLC to *predict* the final loss value. Do this by determining the divisor $L(T)$ that maximally *aligns* the partial curve with the same segment of the corresponding universal TLC. These divisors thus act both as normalizing constants *and*, by construction, as calibrated *extrapolations* of the final loss.
6. Select the optimal hyperparameter setting corresponding to the lowest predicted final loss.

**Predicting the normalized universal TLCs.** In some cases, we may already have many small-scale runs, but not loss curves for each of the *specific* TLC controls corresponding to our large-scale sweep. In this situation, we hypothesize that a *parametric surrogate model* can generate high-fidelity normalized universal TLCs as a function of $\tau$ and TPP. This allows us to obtain the normalized TLC shapes required for Step 5 of the above procedure, *without* training small-models at exactly-matching controls. Fitting such a surrogate lets us leverage our broader TLC dataset, and enables estimation of normalized TLC shapes that go beyond trained regimes (see Appendix D.3 for an example).

For the surrogate $\ell(\hat{t})$ model, we experimented with several functional forms and ablations on our 111M-scale data, focusing on:

$$\hat{\ell}(\hat{t}) = ((1 + \epsilon_1)/(\hat{t} + \epsilon_1))^m + b \cdot (\eta(\hat{t}) + \epsilon_2)^q \tag{4}$$

The first term captures power-law improvement in training fraction (Appendix B.2) while the second term modulates this by the LR schedule $\eta(\hat{t})$, reflecting how variance suppression is phased over training (Appendix B.3). $m$, $b$, $q$, $\epsilon_1$ and $\epsilon_2$ are fit parameters. We divide $\hat{\ell}(\hat{t})$ by its final value so that $\hat{\ell}(1) = 1.0$. Fixing $\epsilon_1 = 0.001$ and $\epsilon_2 = 0.1$ avoids large swings at $\hat{\ell}(0)$ and $\hat{\ell}(1)$.

In practice, we find $m$ can be fixed (we use 0.05). Parameters $b$ and $q$ then vary systematically with $\tau$ and TPP, respectively, which we capture with power laws:

$$b = b_{\text{const}} \cdot (\tau)^{b_{\text{exp}}}, \quad q = q_{\text{const}} \cdot (\text{TPP})^{q_{\text{exp}}} \tag{5}$$

Because $b$ and $q$ interact, jointly fitting their parameters would require a $\mathcal{O}(g^4)$ grid search (with $g$ the grid resolution). Instead, we alternate: fit $(b_{\text{const}}, b_{\text{exp}})$ with fixed $q$, then fit $(q_{\text{const}}, q_{\text{exp}})$ with fixed $b$, iterating to convergence. This reduces cost to $\mathcal{O}(g^2)$ while yielding stable fits.

**Results: prediction.** We fit the $b$ and $q$ power laws on 111M-scale data and evaluate using mean absolute error (MAE) between $\hat{\ell}$ and true $\ell$, computed over $\hat{t} \in [0.2, 1]$ (ignoring error around LR warmup, when initial curves are noisy). We report unweighted mean MAE across all curves.

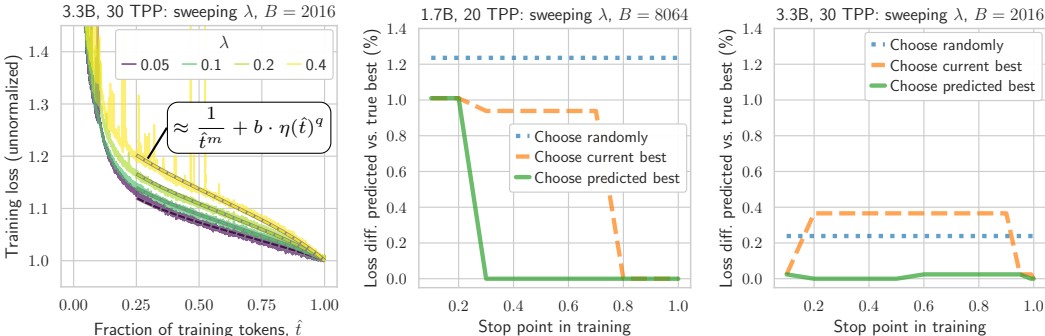

Figure 8: **3.3B-scale predictions and true normalized TLCs.**

Figure 9: **Early stopping works best with *predicted* loss:** Tuning $\lambda$ in 1.7B (*left*) and 3.3B (*right*) models.

Results show that predictions are good: MAE is low and actually improves with scale (appendix Table 11), likely because (1) larger datasets yield smoother TLCs, and (2) fewer extreme hyperparameter tests at larger scales. Fig. 8 (*left*) shows an example: predictions trained on 111M-scale TLCs (1000× fewer FLOPs) closely match observed curves for a 3.3B model.

Estimating $b$ and $q$ as power laws reduces MAE by two-thirds compared to using fixed values (Table 12), though error remains $\approx 2\times$ higher than an oracle fit of $b$ and $q$ *per curve*. Adjusting for both $\tau$ and TPP is vital; however, fitting $b$ and $q$ jointly on both did not improve further.

**Results: tuning.** We now test whether optimal LLM settings can be predicted from partial training runs. At different stopping points in training, we choose a setting as the best, and evaluate the gap between the chosen setting's final loss and the true best setting. We compare the following choices:

1. *Random baseline*: randomly choose one setting as the best.

2. *Current best*: choose the setting giving the best result at the stop point.

3. *Predicted best*: Align partial TLCs with predicted $\hat{\ell}$, choose lowest fitted normalizer $L(T)$.

Fig. 9 shows results for $\lambda$ sweeps at 1.7B/20TPP (*left*) and 3.3B/30TPP (*right*). *Predicted best* achieves negligible loss gaps when stopping after just 30% and 10% of training, respectively. In contrast, *current best*—used in Almazrouei et al. (2023) for LR tuning—succeeds initially at 3.3B but fails at 1.7B, showing it is not a general solution. Further experiments are in Appendix D.2.

> ***Key takeaway 3***: *Collapse enables reliable early stopping. By aligning partial training curves to a small-scale reference, we can predict final loss $L(T)$ and select hyperparameters after only 10–30% of training, substantially reducing tuning compute.*

## 6   RELATED WORK

**Scaling laws and scale-stable dynamics.**   *Neural scaling laws* relate loss (generally obtained from *separate* training runs) to growth in model, data, and compute sizes, via power laws (Hestness et al., 2017; Kaplan et al., 2020; Henighan et al., 2020; Hoffmann et al., 2022; Caballero et al., 2022; Alabdulmohsin et al., 2022). To ensure stable training as models scale, parameterizations such as $\mu$P transfer base hyperparameters across sizes, and yield early dynamics that are scale-stable (Yang et al., 2021; Vyas et al., 2023; Kalra et al., 2023), even *super-consistent* (in curvature) (Noci et al., 2024). Observing *suboptimal LRs* under $\mu$P as *data* scales, recent work has proposed decreasing the LR as a function of $D$ (Shen et al., 2024; Bjorck et al., 2024); Bergsma et al. (2025a) unify these techniques as forms of $\tau$ adjustment. Qiu et al. (2025) show that, for compute-optimal ladders, TLCs collapse after normalization. We build on these threads at LLM scale while co-scaling width, depth, batch size, and weight decay, identifying new controls that govern TLC collapse.

**LLM loss-curve prediction.**   While Kaplan et al. (2020) fit a simple power law to TLCs, recent papers make loss prediction explicitly LR-dependent (Tissue et al., 2024; Luo et al., 2025; Schaipp et al., 2025; Qiu et al., 2025; Hong & Wang, 2025). Complementary to these, we take a timescale-centric view: AdamW implements an EMA over updates, and the normalized timescale $\tau$ (jointly set by LR, weight decay, and batch size) acts to control an *implicit batch size*, one that trades bias reduction vs. variance suppression and thereby shapes TLCs. In a noisy-quadratic model (Appendix B.3), we derive an expression for training loss under a constant LR, and explain why decaying schedules invert the ordering of TLCs across $\tau$, with deformations remaining scale-invariant once normalized.

**Early stopping, HPO, and monitoring.**   Early-termination and HPO methods extrapolate TLCs or prune trials (Swersky et al., 2014; Domhan et al., 2015; Jaderberg et al., 2017; Zela et al., 2018; Li et al., 2018; Choi et al., 2018; Akiba et al., 2019), but typically require many short runs and are not tailored to LLM pre-training regimes. Our approach leverages *collapse itself*: fit a small-scale predictor of normalized TLCs, align in-progress curves to infer $L(T)$, and select winners by 10-30% of training. Operationally, large-scale reports document spikes and divergences (Chowdhery et al., 2022; Zhang et al., 2022a; Wortsman et al., 2023; Molybog et al., 2023); we show collapse residuals provide a quantitative, scale-normalized early-warning signal and a practical aid for debugging.

## 7   CONCLUSION

At LLM scale, *normalized training loss curves collapse across model sizes* when three controls align: the AdamW timescale $\tau$, the tokens-per-parameter ratio (TPP), and the learning-rate schedule. Empirically, $\tau$ (bias–variance smoothing) and TPP (power-law improvement rate) set TLC shape, while the schedule phases these effects. Fixing TPP and setting $\tau$ optimally for that TPP yields alignment across $\sim$100M–3.9B parameters in our experiments.

We instantiate this in **Celerity**: fixed TPP with optimal $\tau$ produces tight collapse and competitive accuracy. *Collapse residuals* surface issues early, localize their onset, and enable safer restarts. A simple predictor for normalized TLCs (fit at small scale) supports *early stopping* in HPO: by 10–30% of training we can select winners and estimate $L(T)$, saving tuning compute. For \$1B runs, collapse provides a valuable reference trajectory: keeping training on track, every step of the way.

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

# A    LIMITATIONS AND FUTURE DIRECTIONS

Across our initial (Secs. 3 and 5) and Celerity (Sec. 4) setups, we have tested collapse across two distinct settings of architecture (and context length), dataset (and vocabulary size), and parameterization. We directly compare TLCs from these two settings in Appendix B.4, while also describing further experiments in learning-rate schedule (*Constant* vs. $10\times$ vs. D2Z), Adam $\beta_1/\beta_2$ parameters, and dense vs. sparse mixture-of-expert (MoE) architectures. However, in all cases our results are established under single-epoch pre-training with AdamW. The observed patterns may change under extreme TPP, multi-epoch training, alternate optimizers/schedules, or heavy mid-training data annealing/curricula.

**Optimizers.**    We hypothesize the optimizer timescale will remain a primary control of TLC shape for other optimizers with decoupled weight decay (e.g., Sophia (Liu et al., 2023), MuonClip (Team, 2025)) whose update rules can be expressed in EMA form analogous to AdamW (Sec. 2). Likewise, the $\tau$ perspective should also hold when AdamW is applied in alternate weight bases, e.g., as in SOAP (Vyas et al., 2024), where AdamW is applied in Shampoo's eigenbasis (Gupta et al., 2018). Extending a timescale analysis to optimizers without a natural EMA form (e.g., Adagrad (Duchi et al., 2011), Adafactor (Shazeer & Stern, 2018), SGD variants) is an important future direction.

**Data curricula.**    Given the growing use of data curricula and late-stage data annealing in LLM pre-training, it is valuable to study how shifts in data affect TLC shape across scales. Collapse may also *inform* curriculum design by serving as a transfer marker. For example, observing limited opportunities for experimentation at large scale, Feng et al. (2024) experiment at smaller scales with *downsampled* datasets that simulate the repetition occurring at larger sizes (due to limited high-quality tokens). Consistency in TLC shape could serve as an indicator of whether the downsample proportions reflect a consistent overfitting/generalization trade-off across scales. Collapse can thus serve to confirm that smaller-scale settings provide suitable proxies for optimizing data mixes and other settings.

**Celerity extensions.**    Beyond choosing TPP (controlling placement on the cost/compression curve; Fig. 5), we aim to understand which factors or training strategies *shift the curve itself*. For faster inference, we are especially interested in the location of the *parameter wall*—the minimal capacity achieving a target loss—and how architecture (dense vs. MoE), routing, and depth/width changes affect collapse and efficiency.

We intentionally chose dense models for our initial Celerity series because dense models have fewer confounding factors (e.g., routing strategy, number of experts), making them simpler to study and build upon. In our own practice, algorithmic innovations are typically validated on dense models first. However, we are interested in scaling MoE-variants of our Celerity series due to their documented savings in training compute (Krajewski et al., 2024; Ludziejewski et al., 2025).

**Train loss vs. generalization.**    We focus on *training* loss because (i) it is FLOPs-free to monitor, (ii) in LLM pre-training it typically tracks validation under stationary data, and (iii) it surfaces issues earlier (e.g., duplicated segments), enabling targeted intervention before held-out degradation. Late-stage annealing and domain shift can decouple train-loss collapse from downstream behavior. We study validation collapse for Celerity in Appendix C.5; future work should also consider *downstream-collapse*, and measure train↔val↔downstream correlations across schedules and data mixtures.

**Predictive model and schedules.**    Both collapse itself, and our predictive model's ability to accurately forecast normalized TLCs, is impaired by loss spikes and divergences, which move the normalized curve away from the universal trajectory (sometimes temporarily, sometimes for extended periods). From one perspective, this is a feature not a bug, as the resulting collapse anomalies provide a useful mechanism for detecting training issues (discussed further below).

Empirically, dividing by the final training loss ($\hat{L}=0$) aligned curves best; future work will study why irreducible-loss offsets, as in Qiu et al. (2025), were not beneficial. In terms of our predictive model, next steps include factoring LR envelope vs. anneal-phase effects (cf. (Tissue et al., 2024; Luo et al., 2025)), adding uncertainty (e.g., seed bootstraps) and uncertainty-aware early-stopping

Table 3: Model architectures used in Sec. 3 and Sec. 5.

| Model | $d_{\text{model}}$ | $n_{\text{layers}}$ | $d_{\text{ffn}}$ | $d_{\text{head}}$ |
|-------|--------|---------|------|-------|
| 111M | 768 | 10 | 2048 | 64 |
| 266M | 768 | 32 | 2048 | 64 |
| 610M | 2048 | 10 | 5461 | 64 |
| 1.7B | 2048 | 32 | 5461 | 64 |
| 3.3B | 2048 | 64 | 5461 | 64 |

policies. Given that in our experiments, the parametric predictor was fit for one specific LR schedule (D2Z), we should also revisit whether $b(\tau)$, $q(\text{TPP})$, and possibly $m$ vary systematically across cosine (Loshchilov & Hutter, 2016), inverse square-root (Vaswani et al., 2017; Raffel et al., 2020; Shen et al., 2024), and warmup-stable-decay (WSD) (Hu et al., 2024; Hägele et al., 2024; Wen et al., 2024; Song et al., 2025) schedules, and schedule-free schemes (Defazio et al., 2024).

It would also be interesting to measure collapse when batch size schedules are used. Theoretically, we could maintain collapse by adjusting weight decay whenever batch size changes, maintaining the $\tau$ invariant. We could compare such adjustments to adjustments of LR, e.g., as in Meterez et al. (2025).

**Systems effects and making collapse a practice.** Collapse residuals are sensitive to systems effects—microbatching/accumulation, precision, kernel changes, restarts—which can create artifacts or reveal true pathologies. To pay down "hidden technical debt" (Sculley et al., 2015), we advocate a lightweight *collapse monitor*: log fraction-of-data in addition to raw step count (easy to add in TensorBoard (Abadi et al., 2016)), as well as microbatch statistics and restart boundaries; normalize online and alert when residuals exceed policy thresholds. Treating collapse as an operational invariant reduces configuration fragility and surfaces data/numerics issues early.

## B EXPLAINING TLC SHAPE: FURTHER DETAILS

### B.1 FULL EXPERIMENTAL DETAILS

In this section, we provide details on the model architecture (Table 3) and training data (Table 4) for models used in experiments in Sec. 3, Sec. 5, and elsewhere in the appendix. Experimental details for the Celerity model series are in Appendix C.2.

In total, ≈600 TLCs were analyzed for these experiments. All such models were GPT2-style LLMs (Radford et al., 2019) with ALiBi (Press et al., 2022) embeddings and SwiGLU (Shazeer, 2020) non-linearity. We use the AdamW optimizer. Following standard practice, we do not apply weight decay or bias to LayerNorm layers. Default AdamW settings are $\beta_1 = 0.9$, $\beta_2 = 0.95$, and $\epsilon = 1\text{e}-8$. We report cross-entropy loss. We parameterize with maximal update parameterization, $\mu$P (Yang et al., 2021), with hyperparameters set via proxy tuning, as described below. For a given TPP, all models have the exact same warmup phase: a linear warmup of the learning rate from 0 to the maximum value. In all runs, warmup was 10% of the total steps. Learning rate warmup is standard practice in LLM training (Brown et al., 2020; Rae et al., 2022; Biderman et al., 2023; Dubey et al., 2024; Kosson et al., 2024).

Note Fig. 3 (and later Table 2, Fig. 7, Fig. 8, and Fig. 9) report batch size in *sequences* rather than *tokens*.

All models in the main experiments were trained on a Cerebras CS-3 system. 610M-parameter 20TPP models take roughly 6 hours each to train on a single CS-3.

**Proxy model hyperparameter tuning.** To find optimal $\mu$P hyperparameters (HPs), we trained a 39M proxy model using a width $d_{\text{proxy}}$ of 256, with 24 layers and head size of 64. We trained this model on 800M tokens with $B$=256 sequences and a context length 2048. We randomly sampled 350 configurations of base learning rates, base initialization standard deviation, and embedding and output logits scaling factors, and used the top-performing values as our tuned HPs (Table 5).

Table 4: Models, tokens-per-parameter (TPP) and corresponding dataset sizes (in tokens), number of model variants trained (over LR schedule type, $\eta$, $\lambda$, $B$) for models used in Sec. 3 and Sec. 5. In total, $\approx$600 TLCs were analyzed.

| Model | TPP | $D$ | Variants trained |
|---|---|---|---|
| 111M | 20 | 2.19B | 74 |
| 111M | 80 | 8.76B | 50 |
| 111M | 200 | 21.9B | 28 |
| 111M | 320 | 35.0B | 40 |
| 111M | 1280 | 140.1B | 11 |
| 266M | 20 | 5.31B | 25 |
| 266M | 80 | 21.2B | 19 |
| 266M | 320 | 85.0B | 19 |
| 266M | 1280 | 339.8B | 3 |
| 610M | 20 | 12.1B | 205 |
| 610M | 80 | 48.5B | 53 |
| 610M | 200 | 121.3B | 14 |
| 610M | 320 | 194.1B | 5 |
| 1.7B | 20 | 34.3B | 31 |
| 1.7B | 80 | 137.2B | 11 |
| 1.7B | 160 | 274.3B | 1 |
| 1.7B | 320 | 548.6B | 1 |
| 3.3B | 20 | 66.5B | 2 |
| 3.3B | 23 | 76.5B | 1 |
| 3.3B | 30 | 99.8B | 5 |

Table 5: Tuned hyperparameters for $\mu$P proxy model for models used in Sec. 3 and Sec. 5.

| | |
|---|---|
| $\sigma_{W,\text{base}}$ | 8.67e-02 |
| $\tilde{\eta}$ | 1.62e-02 |
| $\alpha_{\text{input}}$ | 9.17 |
| $\alpha_{\text{output}}$ | 1.095 |

It is worth noting that the LR values reported in this paper and shown in figures are base $\mu$P LRs before $\mu$P-adjustment. Calculation of $\tau$ (Sec. 2) requires the adjusted LR (i.e., multiplying by $d_{\text{proxy}}/d_{\text{model}}$). Also, when LR decay is used, reported LR values always refer to the peak/max LR of the LR schedule.

## B.2 EXPLAINING TLC DEPENDENCE ON TPP

Schedules with decaying LR reach their minimum value only at the final step (after $D$ tokens). However, for a constant LR schedule, every step of training is equivalent to a complete training run ending at that step. Qiu et al. (2025) make the observation that therefore the loss at every training fraction $\hat{t} = t/T \in [0, 1]$ should respect the same fitted scaling law, but for a training budget of $\hat{t} \cdot D$ tokens.

Starting from the Chinchilla functional form $L(N, D) = E + AN^{-\alpha} + BD^{-\beta}$, assume that we train with a constant LR schedule, training until a certain final tokens-per-parameter ratio $k = D/N$. At every fraction of training $\hat{t}$, we will have trained for an intermediate TPP of $\hat{t} \cdot k$, i.e., using $\hat{t} \cdot k \cdot N$ total tokens. To arrive at scale invariance, we note that Hoffmann et al. (2022) found their fitted model and dataset exponents $\alpha$ and $\beta$ were roughly equal; this rough equality has also been repeatedly validated in replication studies (Besiroglu et al., 2024; Porian et al., 2024). Using $a = \alpha = \beta$, and focusing on the reducible loss, we obtain a final training loss of:

$$L(N, k \cdot N) = AN^{-a} + B(k \cdot N)^{-a}$$
$$= AN^{-a} + Bk^{-a}N^{-a} \tag{6}$$

Meanwhile, training for training fraction $\hat{t}$, the predicted loss is

$$L(N, \hat{t} \cdot k \cdot N) = AN^{-a} + B\hat{t}^{-a}k^{-a}N^{-a} \tag{7}$$

We now normalize by the final loss to expose the shape of the training loss curve. The resulting normalized loss $L(N, \hat{t} \cdot k \cdot N)/L(N, k \cdot N)$ is independent of model (and dataset) size, depending only on the training fraction $\hat{t}$ and the target TPP ratio $k$:

$$\ell(\hat{t}, k) = \frac{A + B\hat{t}^{-a}k^{-a}}{A + Bk^{-a}} \tag{8}$$

In other words, for TLCs using a constant LR schedule, collapse approximately holds under this normalization. Scaling of $\ell$ in $\hat{t}^{-a}$ also motivates our own TLC predictive form (Eq. (4)).

As shown in Appendix C.1, $a = \alpha = \beta$ implies there is a single optimal TPP ratio $r$, and moreover, that the Chinchilla coefficients obey $B = Ar^a$. For a given training run, suppose that the TPP at which we train is a multiple of the optimal TPP by the ratio $v$, e.g., $k = v \cdot r$. Thus, $v = 1$ corresponds to optimal TPP, while $v > 1$ corresponds to overtraining. We can reparameterize the $\ell$ equation in terms of $v$ as:

$$\ell(\hat{t}, v) = \frac{1 + v^{-a}\hat{t}^{-a}}{1 + v^{-a}} \tag{9}$$

This simple equation clarifies how the overtraining factor $v$ influences the shape of the TLCs. When $v$ is small (undertraining), the power law term dominates, and the TLC gradually decays in $\hat{t}$. When $v$ is large (overtraining), the power law only plays a role for smaller $\hat{t}$, the curve drops quickly and then flattens to $\ell = 1$. Intuitively, for overtrained models, we make gains quickly at the beginning of training and then obtain diminishing returns as training progresses.

Qiu et al. (2025) further show that for non-uniform LR schedules, the loss curve is deformed by $\eta(\hat{t})$, but, given consistent curvature of the loss landscape across model scales under $\mu$P (Noci et al., 2024), the noise-induced deformation is invariant to model size, and thus collapse still holds.

## B.3 EXPLAINING TLC DEPENDENCE ON $\tau$

As noted in Sec. 3, the AdamW timescale $\tau = 1/(\eta\lambda T)$ controls the effective memory length of the parameter updates: smaller $\tau$ emphasizes recent updates (bias reduction), while larger $\tau$ averages more broadly (variance reduction). In this sense, $\tau$ acts as an implicit batch size.

To provide further insight into the role of $\tau$ in shaping TLCs, we now derive an analytical expression for training loss under a constant learning rate, using a simple noisy quadratic model (NQM). While LLM training minimizes cross-entropy loss, it is common to perform a local quadratic approximation, i.e., a second-order Taylor expansion in the parameters, with the constant Hessian replaced by the instantaneous Hessian along the training trajectory (LeCun et al., 1989). Thus conclusions drawn from quadratic models often generalize to large, realistic networks (Zhang et al., 2019).

**Setup.** Following Zhang et al. (2019), we assume the optimizer dynamics are invariant to rotation and translation, allowing us to model, without loss of generality, a locally quadratic loss, separable across dimensions, and having an optimum at zero. Specifically, we consider a single quadratic mode with curvature $h > 0$, optimum at $\theta^\star = 0$, and parameters $\theta_t$, where $t$ is the step index:

$$L(t) = \tfrac{1}{2} h \, \theta_t^2. \tag{10}$$

With AdamW optimization, $\theta_t$ evolves as an exponential moving average (EMA) of stochastic updates $x_t$ with constant smoothing $\alpha = \eta\lambda$ (Sec. 3):

$$\theta_t = (1 - \alpha) \, \theta_{t-1} + \alpha \, x_{t-1}. \tag{11}$$

Unrolling the recurrence gives the general form

$$\theta_t \;=\; (1 - \alpha)^t \, \theta_0 \;+\; \sum_{i=0}^{t-1} (1 - \alpha)^{t-1-i} \, \alpha \, x_i. \tag{12}$$

The first term is the (decaying) contribution of the initialization, while the second term reflects the accumulation of stochastic updates.

**Continuous (training-fraction) limit.** We now switch to fractional time $\hat{t} = t/T$ and define the AdamW timescale $\tau = 1/(\alpha T)$. Approximating $(1 - \alpha)^{t-1-i} \approx e^{-\alpha(t-1-i)}$ and interpreting the sum as a Riemann approximation as $T \to \infty$, we obtain

$$\theta(\hat{t}) \;\approx\; e^{-\hat{t}/\tau} \, \theta(0) \;+\; \frac{1}{\tau} \int_0^{\hat{t}} e^{-(\hat{t}-s)/\tau} \, x(s) \, ds. \tag{13}$$

That is, $\theta$ consists of two contributions: a decaying memory of the initialization, and a convolution of the update signal $x(s)$ with an exponential kernel of timescale $\tau$ (an EMA filter over updates).

**Noise model.** Following Zhang et al. (2019), we model the update signal $x(\hat{t})$ as preconditioned white noise: a zero-mean process with constant variance $\sigma_x^2$ and no temporal correlation,

$$\mathbb{E}[x(\hat{t})] = 0, \qquad \mathbb{E}\big[x(\hat{t}) \, x(s)\big] = \sigma_x^2 \, \delta(\hat{t} - s).$$

This idealized assumption isolates the effect of $\tau$ by removing structure in gradient noise beyond its overall scale.

**Mean and variance.** The EMA filter preserves initialization, which decays exponentially:

$$\mathbb{E}[\theta(\hat{t})] \;=\; e^{-\hat{t}/\tau} \, \theta(0).$$

The variance from stochastic updates is

$$\mathrm{Var}[\theta(\hat{t})] \;=\; \frac{\sigma_x^2}{2\tau} \Big( 1 - e^{-2\hat{t}/\tau} \Big). \tag{14}$$

Thus the total second moment is

$$\mathbb{E}[\theta(\hat{t})^2] \;=\; e^{-2\hat{t}/\tau} \, \theta(0)^2 \;+\; \frac{\sigma_x^2}{2\tau} \Big( 1 - e^{-2\hat{t}/\tau} \Big).$$

In words, the initialization bias decays away on timescale $\tau$, while variance from noisy updates accumulates toward a floor proportional to $1/\tau$.

**Expected loss.** The per-mode loss is

$$L(\hat{t}) = \tfrac{1}{2} h\, \theta(\hat{t})^2.$$

Taking expectation, and using the decomposition into bias and variance,

$$\mathbb{E}[L(\hat{t})] \;=\; \tfrac{1}{2} h \left( e^{-2\hat{t}/\tau}\, \theta(0)^2 + \frac{\sigma_x^2}{2\tau} \Big( 1 - e^{-2\hat{t}/\tau} \Big) \right). \tag{15}$$

The first term reflects exponentially decaying initialization bias, while the second reflects variance accumulation to a floor proportional to $1/\tau$.

If initialization is zero-mean in expectation ($\mathbb{E}[\theta(0)^2] = 0$), the bias term vanishes and the expression simplifies to

$$\boxed{\mathbb{E}[L(\hat{t})] \;=\; \frac{h\,\sigma_x^2}{4\tau} \Big( 1 - e^{-2\hat{t}/\tau} \Big)} \tag{16}$$

which captures the characteristic *fast-then-flatten* TLC shape under a constant learning rate.

**Interpretation.** Equation 15 decomposes the expected loss into an exponentially decaying *bias* term ($\propto e^{-2\hat{t}/\tau}\theta(0)^2$) and a *variance* term that rises to a floor ($\propto 1/\tau$). This yields two opposing effects of $\tau$ on TLCs:

- Smaller $\tau$ suppresses initialization bias more rapidly (via the $e^{-2\hat{t}/\tau}$ decay), but accumulates higher variance, yielding a higher asymptotic loss floor ($\propto 1/\tau$).
- Larger $\tau$ reduces variance more effectively, lowering the final loss, but is slower to eliminate bias from initialization.

When initialization is zero-mean in expectation, the bias term vanishes and the expression reduces to Eq. 16.

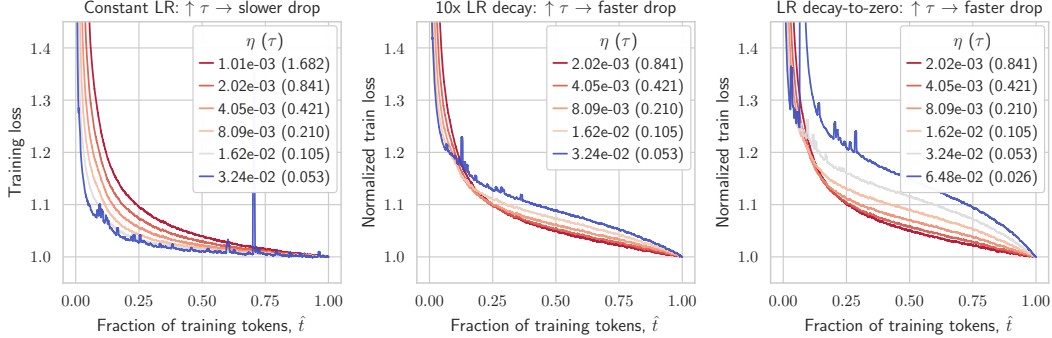

Figure 10: **Effect of LR schedule on TLC shape (610M, 80TPP).** *Left*: *Constant* LR, *Middle*: *Linear* $10\times$ decay, *Right*: *Linear* decay-to-zero. Different schedules deform the TLCs in distinct ways, yet in all cases the AdamW timescale $\tau$ governs the bias-variance trade-off. With a *Constant* LR, smaller $\tau$ accelerates early loss reduction. With D2Z, the effect inverts: smaller $\tau$ yields a larger late-stage drop. Although here $\tau$ is varied by changing LR, equivalent effects arise when varying weight decay or batch size (Fig. 3), confirming $\tau$ as a unifying control knob for TLC shape.

This interpretation matches our empirical findings for normalized *Constant*-LR TLCs (Fig. 10, *left*). The situation is different, however, for LR decay schedules, which we discuss next.

Finally, $\tau$ is a *normalized* timescale and thus invariant to the absolute number of steps. In the NQM, after normalizing by the final loss $L(1)$ the curvature factor $h$ cancels exactly. The normalized curve takes the form

$$\frac{L(\hat{t})}{L(1)} \;=\; \frac{\big(1 - e^{-2\hat{t}/\tau}\big) + \kappa\, e^{-2\hat{t}/\tau}}{\big(1 - e^{-2/\tau}\big) + \kappa\, e^{-2/\tau}}, \qquad \kappa \;=\; \frac{2\tau\,\mathbb{E}[\theta(0)^2]}{\sigma_x^2}.$$

Thus, when the initialization contribution is negligible by the end of training (or when the ratio $\kappa$ is approximately scale-invariant), the normalized TLC depends only on $(\tau, \hat{t})$, and curves at matched

$\tau$ collapse across model sizes. If $\kappa$ varies across scales, small early deviations can appear (bias-dominated regime) but typically diminish as $e^{-2\hat{t}/\tau}$ decays.

*Remark.* Qiu et al. (2025) observed collapse without AdamW. Empirically, as $\lambda \to 0$, TLCs approach a limiting shape: vanilla Adam behaves like AdamW with $\lambda = 0$ (effectively $\tau = \infty$).

**Extension to decaying LR schedules.** The constant-LR analysis in Eq. 16 shows that $\tau$ sets the trade-off: smaller $\tau$ accelerates early bias reduction but saturates at a higher variance-driven floor, while larger $\tau$ reduces variance more slowly but to a lower asymptote. With a decaying LR schedule, the smoothing $\alpha_t = \eta_t \lambda$ *decreases* after warmup, so the instantaneous timescale $\tau_t = 1/(\eta_t \lambda T)$ *increases* as training progresses. In this setting, small-$\tau$ runs still make rapid early progress (fast bias reduction), but during the decay phase they gain additional variance suppression as $\tau_t$ lengthens, often producing a noticeable late-stage drop in loss. By contrast, large-$\tau$ runs emphasize variance reduction throughout, yielding steadier curves without the same end-of-training acceleration. Equivalently, in the EMA view, decay flattens the contribution coefficients $c_{t,i}$, averaging over more (earlier) updates near the end. Thus LR decay effectively combines the early bias-reducing dynamics of small $\tau$ with the late variance-reducing dynamics of large $\tau$, inverting the TLC ordering observed under constant LR (Fig. 10).

This analysis aligns with Bergsma et al. (2025b), who attribute the effectiveness of D2Z schedules to balancing early bias reduction with later variance suppression (building on D'Angelo et al., 2024). Their treatment is primarily conceptual; here we show how the same bias–variance dynamics manifest directly in TLC shapes and provide a simple analytical form under the NQM.

## B.4 ADDITIONAL TLC EXPERIMENTS

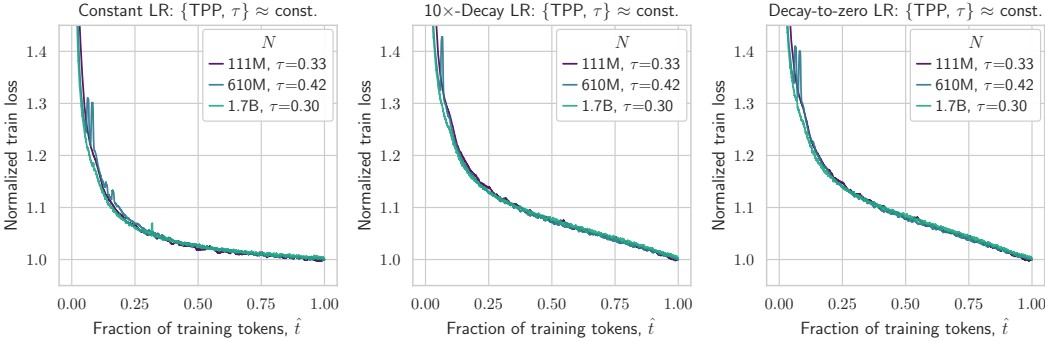

Figure 11: **Collapse in different LR schedules.** *Left*: *Constant* LR, *Middle*: *Linear* $10\times$ decay, *Right*: *Linear* decay-to-zero. In contrast to Fig. 10, where $\tau$ varies, here TPP=20 and $\tau \approx 0.3$: curves collapse across scales.

**Collapse under alternative LR schedules.** Fig. 11 shows that normalized TLCs also collapse under a *Constant* schedule, a $10\times$ decay schedule, and our decay-to-zero schedule (all with 10% warmup). At corresponding model sizes, we use the same batch size, peak LR, and weight decay, so same-size results across schedules differ only in their final LR. Collapse is slightly looser than in the Celerity runs because the resulting $\tau$ is not matched exactly across schedules (see plot annotations), but the qualitative agreement is strong. These results are consistent with our analysis in Appendix B.3 and echo the cross-schedule findings of Qiu et al. (2025).

**Collapse across datasets and architectures.** TLC shape can in principle depend on task, data, and architecture (e.g., multi-epoch training on a small corpus can yield faster apparent improvement than single-epoch pre-training). We therefore ask: how much does normalized TLC shape change as we vary parameterization, vocabulary size, architecture, context length, and dataset mix?

As a first probe, we compare *Celerity* TLCs to our earlier non-Celerity runs, at the same TPP (20) and similar $\tau \approx 0.2$, while varying all items above: Celerity uses CompleteP (vs. vanilla $\mu$P), a larger vocabulary, different nonlinearity and FFN multiplier, $4\times$ longer context, and a different data

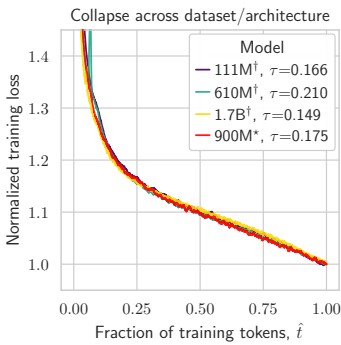
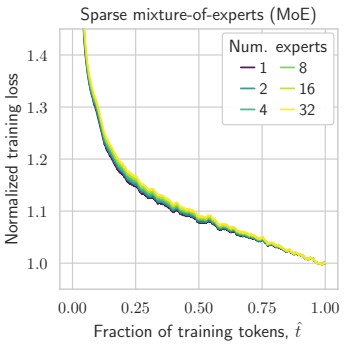
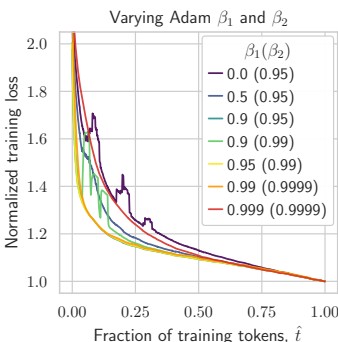

Figure 12: **Collapse across $^\dagger$original, $^\star$Celerity setups.** 20 TPP.

Figure 13: **Collapse as $E$ varies in a sparse MoE.** 111M, $\tau = 0.33$, 20 TPP.

Figure 14: **Collapse as Adam $\beta_1$ and $\beta_2$ vary.** 610M, $\tau = 0.21$, 20 TPP.

mixture (Appendices B.1 and C.2). Despite these differences, the TLCs loosely collapse (Fig. 12). The Celerity 900M model tracks closer to the 610M model than to the 1.7B model, although its $\tau$ is intermediate between these two. Overall, we view this as evidence that the normalized TLC shape is surprisingly robust when LR schedule, TPP, and $\tau$ are held (approximately) fixed.

**Collapse in sparse mixture-of-experts (MoE).** We next analyze sparse MoE architectures, where only a subset of parameters are active per token (Lepikhin et al., 2020; Fedus et al., 2022). Starting from our 111M dense model (Appendix B.1), we replace each FFN with a sparse MoE layer and vary the number of experts $E \in \{1, 2, 4, 8, 16, 32\}$. Tokens are routed to one expert via hash routing (Roller et al., 2021) so each expert processes a similar token count. Global training tokens and datasets are identical across $E$, hence the *effective* TPP per expert decreases from 20 (dense) to $20/E$ as $E$ grows. Note also that as the number of experts $E$ increases, and the effective tokens per expert decrease proportionally, both the expert's effective batch size $B$ and effective dataset size $D$ are reduced by a factor of $E$. Since $\tau = B/(\eta\lambda D)$, these reductions cancel, leaving the overall timescale unchanged (for fixed $\eta, \lambda$).

Fig. 13 shows that lower $E$ (higher effective TPP per expert) yields slightly earlier drops and slightly flatter tails, broadly obeying the TPP effect characterized in Sec. 3. Thus, the observed deformation is explained by effective TPP rather than differing training dynamics per se.

**Collapse across Adam $\beta_1$ and $\beta_2$.** Finally, we vary $(\beta_1, \beta_2)$ at fixed LR, batch size, and weight decay ($\tau = 0.21$, 610M model, 20 TPP). The default $(0.9, 0.95)$ gives the lowest absolute loss in this experiment, but several "standard" settings—$(0.9, 0.95)$, $(0.95, 0.99)$, and even $(0.99, 0.9999)$—produce normalized TLCs that collapse (Fig. 14). In contrast, runs with $(0.0, 0.95)$, $(0.5, 0.95)$, and a noisy instance of $(0.9, 0.99)$ exhibit early loss spikes; when the loss fails to recover promptly, the curves remain elevated and do not rejoin the main trajectory, breaking collapse (early loss spikes also distort early collapse for noisy, large-batch-size runs, e.g., Fig. 7). We also observe that $(0.999, 0.9999)$, which aggregates gradients over a much longer horizon, follows a systematically slower (but eventually convergent) trajectory—consistent with an enlarged momentum timescale prioritizing variance reduction over bias, akin to increasing $\tau$.

Overall, aside from extreme momentum settings or instability-induced spikes, setting of $(\beta_1, \beta_2)$ has limited effect on the *shape* of normalized TLCs. The AdamW timescale $\tau$ remains the dominant optimization-based control for TLC trajectories.

> *Key takeaway 4: Normalized TLCs are strikingly robust: they largely collapse across diverse datasets and architectures, remain predictable under sparse MoE routing (scaling in effective TPP as theory suggests), and are insensitive to typical Adam $\beta_1, \beta_2$ settings. Apart from pathological loss spikes, the dominant factor shaping TLCs is still the AdamW timescale $\tau$.*

## C CELERITY MODELS: FURTHER DETAILS

### C.1 COMPUTE COST AS A FUNCTION OF MODEL COMPRESSION

Starting from a compute-optimal model size, we now derive an expression for the extra compute required ($C/C_{\text{opt}}$) to compress a model to a smaller (less efficient) size, while maintaining the *same loss*. We use the resulting equation to plot the compression vs. cost trade-off in Fig. 5. This analysis motivated the selection of max TPP in the Celerity model series.

We begin again with the Chinchilla functional form from Hoffmann et al. (2022), giving loss $L$ as a function of model size $N$ and data size $D$:

$$L(N, D) = E + AN^{-\alpha} + BD^{-\beta} \tag{17}$$

where $E$, $A$, $\alpha$, $B$, and $\beta$ are parameters to be fit on observed training runs.

Hoffmann et al. (2022) asked, for a fixed training compute budget $C$ (in FLOPs), how should we allocate model size $N$ versus number of training tokens $D$ in order to minimize *loss*? From Eq. (17), they derived functions for loss-optimal $N_{\text{opt}}(C)$ and $D_{\text{opt}}(C)$ (constraining $L(N, D)$ by the common approximation $C \approx 6ND$):

$$N_{\text{opt}}(C) = G\left(\frac{C}{6}\right)^{\frac{\beta}{\alpha+\beta}} \text{ and } D_{\text{opt}}(C) = G^{-1}\left(\frac{C}{6}\right)^{\frac{\alpha}{\alpha+\beta}}, \tag{18}$$

where $G = \left(\frac{\alpha A}{\beta B}\right)^{\frac{1}{\alpha+\beta}}$. Results indicated that $N_{\text{opt}}$ and $D_{\text{opt}}$ scale roughly equally as $C$ increases. This analysis agreed with their other methods for estimating compute-optimal scaling, and guided their $N$ and $D$ allocation for training their large-scale Chinchilla model.

*Let $r$ be the optimal $D_{\text{opt}}(C)/N_{\text{opt}}(C)$ ratio.* If $r$ is roughly independent of $C$, this implies $\alpha \approx \beta$. Using $a = \alpha = \beta$, we obtain:

$$r = \left(\frac{B}{A}\right)^{\frac{1}{a}}, \tag{19}$$

or equivalently $B = Ar^a$.

Replication studies have found $\alpha \approx \beta \approx 0.35$, and an optimal TPP of around $r = 20$ (Besiroglu et al., 2024; Porian et al., 2024) (as noted in Sec. 2).

Now, suppose $a = \alpha = \beta$ and we obtain a loss of $\hat{L}$ at the optimal TPP ratio (where $D_{\text{opt}} = rN_{\text{opt}}$):

$$\begin{aligned}
\hat{L} &= E + AN_{\text{opt}}^{-\alpha} + BD_{\text{opt}}^{-\beta} \\
&= E + AN_{\text{opt}}^{-a} + (Ar^a)(rN_{\text{opt}})^{-a} \\
&= E + 2AN_{\text{opt}}^{-a}
\end{aligned} \tag{20}$$

We now wish to train a *compressed* model with fraction $k_N$ of parameters compared to $N_{\text{opt}}$, but obtaining the same loss. Let $N = k_N N_{\text{opt}}$. If $N < N_{\text{opt}}$, we will need $k_D$ extra tokens compared to $D_{\text{opt}}$ in order to reach the loss target. Let $D = k_D D_{\text{opt}}$. Rather than training at $r$ TPP, we will train at a higher ratio $(k_D D_{\text{opt}})/(k_N N_{\text{opt}}) = (k_D/k_N)r$. From Eq. (17), and following a similar derivation to De Vries (2023), the estimated loss will be:

$$L(N, D) = E + A(k_N N_{\text{opt}})^{-\alpha} + B(k_D D_{\text{opt}})^{-\beta} \tag{21}$$

Again substituting $a = \alpha = \beta$ and $B = Ar^a$, to obtain the target loss $\hat{L}$, we set the loss in Eq. (21) to equal $\hat{L}$ in Eq. (20), and solve for $k_D$, finding:

$$k_D = \left(2 - k_N^{-a}\right)^{\frac{-1}{a}} \tag{22}$$

The compute cost $C$ of the compressed training will be $6(k_N N_{\text{opt}})(k_D D_{\text{opt}})$, from which we can derive the extra compute ratio compared to $C_{\text{opt}} = 6N_{\text{opt}} D_{\text{opt}}$:

$$\begin{aligned}
C/C_{\text{opt}} &= k_N k_D \\
&= k_N \left(2 - k_N^{-a}\right)^{\frac{-1}{a}}
\end{aligned} \tag{23}$$

Eq. (23) allows us to vary $k_N$ and obtain the corresponding compute overhead. When planning the Celerity training runs, we assumed $r = 20$ corresponded to the compute-optimal model size (following the Chinchilla rule-of-thumb) and we tested different values of $a$ reported in prior work, using $a = 0.35$ in Fig. 5.

## C.2 CELERITY RECIPE DETAILS

In this section, we provide further details for the techniques that most impacted Celerity's performance and compute efficiency, including parameterization, learning rate and weight decay scheduling, architecture, and dataset construction.

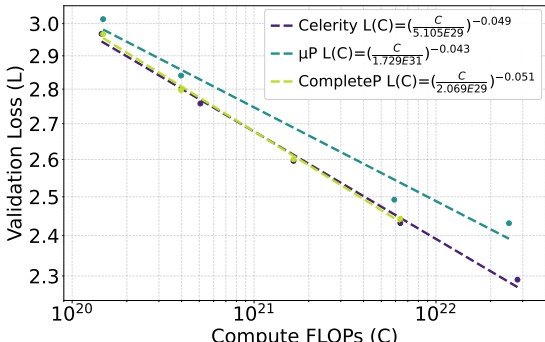

Figure 15: Scaling law comparison between CompleteP and $\mu$P. CompleteP scales more predictably: the power law is a better fit to the observed points. CompleteP also exhibits a steeper slope, improving loss faster in compute FLOPs, likely due to both better HP transfer (across model width *and* depth), and better compute efficiency.

**Parameterization.** We compare the effect of different parameterizations and their influence on compute efficiency in Fig. 15. Specifically, we compare $\mu$P (Yang et al., 2021), which accounts for scaling in *width*, and CompleteP (Dey et al., 2025), which accounts for scaling in *both width and depth*. For each parameterization, the hyperparameters (HP), such as learning rate, weight initialization, and multipliers, are tuned at depth 32 and then directly applied to the target model training.

Here we note two observations: the first observation is that $\mu$P points do not align well on a standard scaling law. We attribute this to HP de-tuning when transferring HPs from proxy model depth to target model depth. Such de-tuning is not seen in the scaling laws for CompleteP and Celerity (which uses CompleteP but a different dataset), where the points align with minimal error on the scaling law. Dey et al. (2023a) showed poor scaling law fits when scaling width in the *standard parameterization* vs. $\mu$P; we believe a similar phenomenon is now happening when scaling in *depth*.

The second observation is that CompleteP is more compute efficient than $\mu$P, which prior work has explained through the lens of feature learning (Dey et al., 2025). Indeed, Fig. 4 in Dey et al. (2025) suggest that CompleteP models exhibit better scaling behavior than $\mu$P, even when both are comprehensively tuned.

Based on these observations, we used CompleteP for training the Celerity series, using the proxy model's tuned HPs across all scales. Sample code for implementing CompleteP is available at https://github.com/EleutherAI/nanoGPT-mup/tree/completep.

**Learning Rate and Weight Decay.** We chose the linear decay-to-zero (D2Z) learning rate schedule based on its empirical success and conceptual motivations in Bergsma et al. (2025b). In particular, Bergsma et al. (2025b) showed that as TPP increases beyond compute-optimal 20 TPP, the relative benefit of D2Z also increases, in a scale-invariant manner. This makes D2Z particularly appropriate for parameter-efficient training (e.g., Celerity's 234 TPP model band). All models also train with linear warmup to the peak LR, over the minimum of 10%-of-total-tokens or 375M-tokens.

We tuned $\tau$ at a smaller scale and smaller TPP, and transferred across TPP using the power law fit from Bergsma et al. (2025a). Given learning rate is determined by CompleteP, and batch size is optimized according to a separate scaling rule (described below), we adjusted weight decay in order to obtain the desired $\tau$ setting at each scale.

**Batch Size.** In early experimentation, the batch sizes were chosen such that they were around the critical batch size (McCandlish et al., 2018). Later we used the insights from Bergsma et al. (2025a) and started following the rule $B_{\text{opt}} \propto D^{0.5}$, tuning $B$ at a small scale and then inferring optimal batch sizes on larger datasets via the power law.

Table 6: Composition of the Celerity pre-training dataset.

| Data Subset | Percentage (%) |
|---|---|
| FineWeb-Edu (Lozhkov et al., 2024) | 64.75 |
| StarCoder (Li et al., 2023) | 10.8 |
| Cosmopedia (Ben Allal et al., 2024) | 4.66 |
| SlimPajama (Soboleva et al., 2023) | 17.49 |
| OpenWebMath (Paster et al., 2023) | 1.88 |
| UltraTextBooks-2.0 (Gabarain, 2024) | 0.42 |

**Data Selection.** Over the course of experiments, we found that adding more *refined* data, particularly educational, math, and coding datasets, generally helps the models score higher on common benchmarks. In Table 6, we break down the datasets used for Celerity model training, including the proportion assigned to each subset. A large portion of the datasets are focused on educational materials, math, and coding. We use only the *non–web-crawled* subsets of SlimPajama (Soboleva et al., 2023), i.e., excluding C4 and CommonCrawl, which are effectively replaced by FineWeb-Edu. As noted in Sec. 4, we do not schedule the data sources, i.e., we do not employ a data curriculum in the training of Celerity, nor do we include (benchmark) task-specific data in Celerity training.

Table 7: Comparison of Celerity models trained on different datasets.

| Name | Downstream Accuracy (Num Shots) | | | | | | | |
|---|---|---|---|---|---|---|---|---|
| | arc-c (25) | arc-e (0) | boolq (0) | hellaswag (10) | piqa (0) | siqa (0) | winogrande (5) | Avg. |
| Celerity 300M | 27.82 | 50.63 | 52.75 | 37.57 | 66.21 | 37.77 | 52.25 | **46.43** |
| Celerity 300M SlimPJ | 24.32 | 42.17 | 61.53 | 36.04 | 65.56 | 37.97 | 50.99 | 45.51 |
| Celerity 900M | 39.68 | 64.52 | 47.92 | 55.02 | 72.03 | 41.97 | 58.48 | **54.23** |
| Celerity 900M SlimPJ | 30.89 | 54.67 | 55.47 | 53.74 | 71.00 | 40.89 | 57.46 | 52.02 |

Comparison in Table 7 shows that the same model configurations trained on a general dataset like SlimPajama result in worse downstream performance compared to the Celerity data mix. While dataset optimization was not a focus of Celerity, these results do underscore the importance of dataset composition in pre-training. This also makes clear why hyperscalers invest a tremendous amount of work into data preparation, synthesis, filtering, and refinement.

Table 8 summarizes the dataset sizes for all models in the Celerity model series.

**Model Architecture.** Celerity models use a decoder-only GPT2-style transformer architecture. Table 2 summarizes the architecture dimensions, hyperparameters, and other details of the Celerity model family. We trained five Celerity model sizes from scratch with parameter counts roughly 300M, 500M, 900M, 1.8B, and 3.9B. All models are trained under consistent data and optimization methods, on public datasets, in order to foster open science and fair comparison.

## C.3 CELERITY FURTHER RESULTS

In our empirical evaluation of Celerity, we necessarily only compare to model families with sufficiently precise and complete training details, in particular the total training tokens. E.g., Llama-3.2

Table 8: Models, tokens-per-parameter and corresponding dataset sizes (in tokens) for Celerity.

| Model | TPP | $D$ |
|---|---|---|
| 300M | 20 | 5.4B |
| 300M | 80 | 21.7B |
| 300M | 234 | 63.4B |
| 500M | 20 | 10.1B |
| 500M | 80 | 40.2B |
| 500M | 234 | 117.8B |
| 900M | 20 | 18.1B |
| 900M | 80 | 72.5B |
| 900M | 234 | 212.3B |
| 1.8B | 20 | 36.2B |
| 1.8B | 80 | 144.8B |
| 1.8B | 234 | 424.0B |
| 3.9B | 20 | 77.6B |
| 3.9B | 80 | 310.4B |
| 3.9B | 234 | 909.2B |

reports using "up to" 9T tokens (Llama-3.2 Overview) while Llama-3.1 and Qwen-3 sizes are reported "approximately." Moreover, it is unclear from the papers whether the full pre-training datasets (i.e., used to train the flagship models) were also used for the smaller models. For an approximate sense of how these models compare, we include Llama-3 and Qwen models in Table 10 based on the assumption the models do use the full (approximately-reported) datasets. Note that if these assumptions hold, the smaller Llama-3 and Qwen-3 models would be rather compute-inefficient, e.g., they would all be beyond the plotting range of Fig. 2 (i.e., $> 10^{24}$ FLOPs) and score well below the Celerity extrapolation (and Gemma results) in Fig. 16.

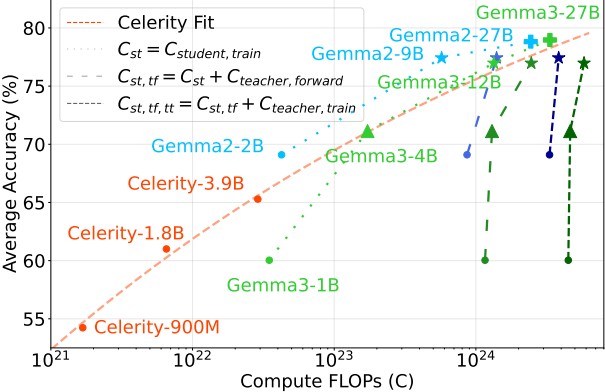

Figure 16: **Celerity compute efficiency vs. distilled models:** Downstream accuracy. Celerity models perform similarly to *distilled* Gemma-2/Gemma-3 models, when generously only accounting for distillation student FLOPs. When considering *teacher* forward pass FLOPs, Gemma curves shift away from Pareto frontier (worse), with a further shift if we account for FLOPs to *train* the teacher.

**Compute efficiency.** Fig. 16 provides further downstream results for Celerity models (and their fitted extrapolation), in comparison to larger Gemma-2 and Gemma-3 models. The plot shows how the accuracy vs. FLOPs comparison depends on whether we account for teaching FLOPs (e.g., generating logits for student training), or the initial cost of educating the teacher.

**Token efficiency.** Fig. 17 compares the *token* efficiency of Celerity to other model families. Using the Celerity models trained in the fixed 234 TPP band, we fit a power law in $D$ and extrapolate token efficiency to larger scales.

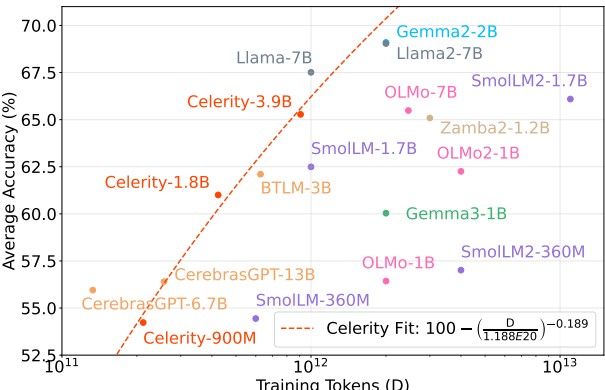

Figure 17: **Celerity token efficiency:** Downstream accuracy. Celerity models are at the Pareto frontier compared to other model families.

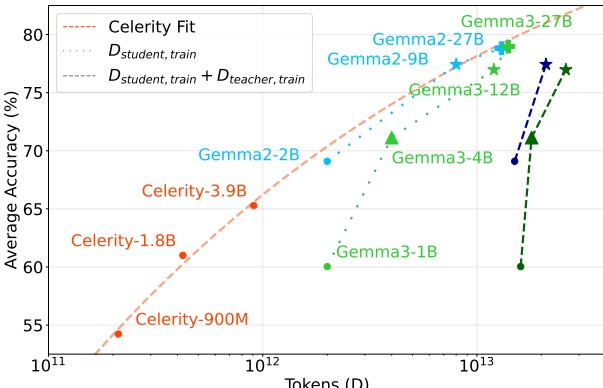

Figure 18: **Celerity token efficiency vs. distilled models:** Downstream accuracy. Celerity models are on par with distilled models.

Generally, larger models should be more token-efficient for the same token budget. Theoretically, distillation should also offer greater token efficiency—at a given TPP (Busbridge et al., 2025)—but by training small models to very-high TPP, the distilled models in Fig. 18 train mainly in a regime of diminishing returns, and so ultimately end up without an advantage over Celerity's standard next-token-prediction training.

There are many interesting questions around token efficiency at scale, and indeed token efficiency may become more critical as frontier models reach the limits of high-quality data (Muennighoff et al., 2023).

**Parameter efficiency.**  Finally, Figs. 19 and 20 provide the parameter efficiency comparisons for Celerity. Celerity models are less parameter efficient than models specifically designed for parameter efficiency.

### C.4  OPEN MODEL EVALUATION AND FLOP CALCULATION METHODS

All models are obtained from HuggingFace and evaluated using the Eleuther Eval Harness framework (Gao et al., 2021). The downstream tasks with number of shots are arc-challenge (25), arc-easy (0), boolq (0), hellaswag (10), piqa (0), siqa (0) and winogrande (5). These tasks are chosen as they are the most commonly reported downstream benchmarks for pre-trained base models, and are appropriate for Celerity models of the scale that we compare (i.e., tasks where small models perform above random chance).

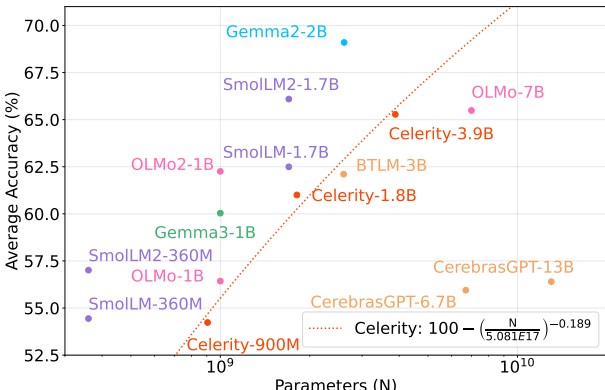

Figure 19: **Celerity parameter efficiency:** Downstream accuracy. Celerity models are less parameter efficient than models trained at much higher TPP, while better than prior models also aiming for compute efficiency (Cerebras GPT).

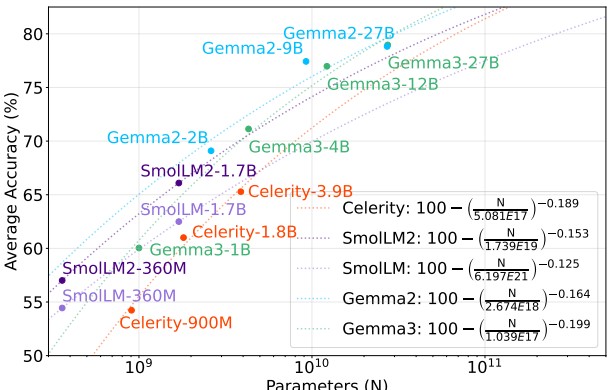

Figure 20: **Celerity parameter efficiency scaling comparison:** Downstream accuracy. Preliminary accuracy vs. model size power law comparison between Gemma (distillation), SmolLM (refined data), and Celerity models (standard pre-training). Distilled model families have the largest scaling exponent, suggesting distillation may scale better in parameters.

For full transparency, our method for counting FLOPs across the different models families is given in Table 9, while a table of all the raw downstream evaluation scores are in Table 10.

## C.5 VALIDATION LOSS COLLAPSE

In Fig. 21 we show normalized training loss curves where we evaluate the *validation loss* of model checkpoints during training. For each training run, we evaluate checkpoints at 5% intervals, computing loss on the same 493M-token held-out portion of SlimPajama. Similar to training, collapse occurs except for the initial few checkpoints; we attribute the initial differences to differing LR warmup proportions (Table 2). Nevertheless, validation collapse is sufficient for deviations to provide a useful diagnostic of any training issues. Validation collapse, measured on fixed datasets, could be a particularly valuable diagnostic if late-stage annealing or curriculum strategies distort training curves due to data differences.

Table 9: Forward FLOPs calculation for self-attention block and Mamba-2 block. This table only lists operations that are not covered $6 * n_{params} * n_{tokens}$, which should take care of all operations that involves a matmul with a weight matrix. For Zamba2, the training FLOPs can be calculated as $6 * n_{params} * n_{tokens} - 2 * V * D_{attn} * n_{tokens} + 3 * L_{attn} * (L * C_{mamba2} + C_{attn})$, while the rest of the models analyzed are variations of decoder-transformers whose training FLOPs can be estimated as $6 * n_{params} * n_{tokens} - 2 * V * D_{attn} * n_{tokens} + 3 * L_{attn} * C_{attn}$. Here 3 represents 1 flop per forward op and 2 flops per backward op.

|  | Operation | FLOPs, given input: $B \times S \times D$ (or $D_{attn}$) |
|---|---|---|
| Self-Attention $C_{attn}$ | Attention: $QK^T$ | $2BS^2D_{attn}$ |
|  | Attention: softmax, scaling, mask | $3BS^2$ |
|  | Attention: $V$ matmul | $2BS^2D_{attn}$ |
|  | Attention: $O$ projection | $2BSD_{attn}^2$ |
|  | Feedforward: activation | $BSD_{attn}$ |
| Mamba-2 $C_{mamba2}$ | dt softplus | $3BSH$ |
|  | xBC conv1d, silu | $BS(ED+2N)K + 5BS(ED+2N)$ |
|  | sampling x, A | $BSED + BSH$ |
|  | SSD, A prefix sum | $BHS$ |
|  | SSD, compute output for each intra-chunk | $4BHSC + BSEDNC$ |
|  | SSD, compute state for each intra-chunk | $2BHS + BSEDN$ |
|  | SSD, compute inter-chunk recurrence | $4BH(Z+1)^2 + BN(Z+1)^2ED$ |
|  | SSD, compute output from state per chunk | $BHS + BSEDN + BSED$ |
|  | y+x*D | $2BSED$ |
|  | z silu, y norm | $6BSED$ |
| Params | $B$: batch size, $S$: sequence length, $V$: vocabulary size $D_{attn}$: attention hidden dim, $L_{attn}$: num attention layers $D$: mamba2 hidden dim, $L$: num mamba2 layers, $E$: expansion factor $N$: mamba2 state dim, $H$: mamba2 num heads, $P$: mamba2 head dim $C$: mamba2 chunk size, $Z$: mamba2 num chunks, $K$: mamba2 conv dim | |

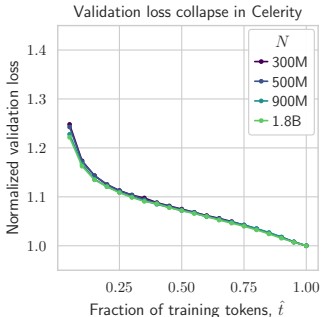

Figure 21: **Training loss curves also collapse in *validation* loss:** Normalized validation loss across 4 Celerity model sizes, all trained to 80 TPP. Validation loss collected at 5% intervals of training.

## D    COLLAPSE ENABLES EARLY STOPPING: FURTHER DETAILS

### D.1    PREDICTING NORMALIZED TRAINING LOSS CURVES

This section provides further details regarding the development of the functional form in Eq. (4), which we use to predict normalized TLCs and, through these, extrapolate in-progress TLCs. Based on Sec. 3, we know that TLC shape is modulated by LR schedule, TPP, and $\tau$. Prior theoretical and empirical work has mostly focused on how loss proceeds as a function of training steps and LR schedule (Defazio et al., 2023; Tissue et al., 2024; Schaipp et al., 2025; Luo et al., 2025; Qiu et al., 2025). To incorporate these factors into a single functional form, we take the following approach:

Table 10: Evaluations, params, tokens, and FLOPs for all models evaluated.

| Name | Params | Tokens | FLOPs | Downstream Accuracy (Num Shots) | | | | | | | |
| | | | | arc-c (25) | arc-e (0) | boolq (0) | hellaswag (10) | piqa (0) | siqa (0) | winogrande (5) | Avg. |
|---|---|---|---|---|---|---|---|---|---|---|---|
| BTLM-3b-8k-base (Dey et al., 2023b) | 2.60E+09 | 6.27E+11 | 1.55E+22 | 40.70 | 66.79 | 69.72 | 70.92 | 77.20 | 43.50 | 65.90 | 62.10 |
| Cerebras-GPT-1.3B (Dey et al., 2023a) | 1.30E+09 | 2.60E+10 | 2.45E+21 | 26.79 | 45.83 | 59.33 | 38.55 | 66.76 | 38.59 | 51.70 | 46.79 |
| Cerebras-GPT-2.7B | 2.70E+09 | 5.40E+10 | 1.04E+21 | 29.52 | 52.57 | 59.24 | 49.74 | 70.78 | 40.23 | 54.85 | 50.99 |
| Cerebras-GPT-6.7B | 6.70E+09 | 1.33E+11 | 6.16E+21 | 36.01 | 57.91 | 62.81 | 59.45 | 73.99 | 41.50 | 59.98 | 55.95 |
| Gemma-2-2b (Team et al., 2024) | 2.61E+09 | 2.00E+12 | 4.25E+22 | 53.41 | 80.22 | 73.58 | 74.62 | 79.11 | 51.23 | 71.51 | 69.10 |
| Gemma-2-9b | 9.24E+09 | 8.00E+12 | 5.73E+23 | 68.34 | 87.88 | 84.22 | 82.76 | 82.97 | 55.48 | 80.35 | 77.43 |
| Gemma-2-27b | 2.72E+10 | 1.30E+13 | 2.44E+24 | 69.62 | 88.30 | 84.83 | 87.00 | 84.44 | 54.55 | 83.03 | 78.82 |
| Gemma-2-2b+forward | 2.61E+09 | 2.00E+12 | 8.66E+23 | 53.41 | 80.22 | 73.58 | 74.62 | 79.11 | 51.23 | 71.51 | 69.10 |
| Gemma-2-2b+forward+teacher | 2.61E+09 | 1.50E+13 | 3.31E+24 | 53.41 | 80.22 | 73.58 | 74.62 | 79.11 | 51.23 | 71.51 | 69.10 |
| Gemma-2-9b+forward | 9.24E+09 | 8.00E+12 | 1.40E+24 | 68.34 | 87.88 | 84.22 | 82.76 | 82.97 | 55.48 | 80.35 | 77.43 |
| Gemma-2-9b+forward+teacher | 9.24E+09 | 2.10E+13 | 3.84E+24 | 68.34 | 87.88 | 84.22 | 82.76 | 82.97 | 55.48 | 80.35 | 77.43 |
| Gemma-3-1b-pt (Team et al., 2025) | 1.00E+09 | 2.00E+12 | 3.48E+22 | 39.16 | 71.93 | 66.67 | 62.98 | 74.54 | 42.78 | 62.19 | 60.04 |
| Gemma-3-4b-pt | 4.30E+09 | 4.00E+12 | 1.72E+23 | 58.28 | 81.69 | 78.96 | 77.78 | 79.87 | 49.13 | 72.22 | 71.13 |
| Gemma-3-12b-pt | 1.22E+10 | 1.20E+13 | 1.34E+24 | 67.49 | 87.75 | 85.41 | 84.12 | 81.88 | 52.15 | 80.03 | 76.98 |
| Gemma-3-27b-pt | 2.74E+10 | 1.40E+13 | 3.33E+24 | 70.31 | 88.17 | 87.25 | 86.14 | 83.95 | 53.99 | 82.95 | 78.97 |
| Gemma-3-1b-pt+forward | 1.00E+09 | 2.00E+12 | 1.16E+24 | 39.16 | 71.93 | 66.67 | 62.98 | 74.54 | 42.78 | 62.19 | 60.04 |
| Gemma-3-1b-pt+forward+teacher | 1.00E+09 | 1.60E+13 | 4.49E+24 | 39.16 | 71.93 | 66.67 | 62.98 | 74.54 | 42.78 | 62.19 | 60.04 |
| Gemma-3-4b-pt+forward | 4.00E+09 | 4.00E+12 | 1.30E+24 | 58.28 | 81.69 | 78.96 | 77.78 | 79.87 | 49.13 | 72.22 | 71.13 |
| Gemma-3-4b-pt+forward+teacher | 4.00E+09 | 1.80E+13 | 4.63E+24 | 58.28 | 81.69 | 78.96 | 77.78 | 79.87 | 49.13 | 72.22 | 71.13 |
| Gemma-3-12b-pt+forward | 1.20E+10 | 1.20E+13 | 2.46E+24 | 67.49 | 87.75 | 85.41 | 84.12 | 81.88 | 52.15 | 80.03 | 76.98 |
| Gemma-3-12b-pt+forward+teacher | 1.20E+10 | 2.60E+13 | 5.80E+24 | 67.49 | 87.75 | 85.41 | 84.12 | 81.88 | 52.15 | 80.03 | 76.98 |
| Llama-7b (Touvron et al., 2023a) | 7.00E+09 | 1.00E+12 | 4.82E+22 | 50.77 | 72.90 | 75.05 | 77.84 | 79.00 | 45.91 | 71.11 | 67.51 |
| Llama-13b | 1.30E+10 | 1.00E+12 | 8.90E+22 | 55.55 | 74.54 | 77.98 | 81.18 | 80.36 | 46.62 | 76.95 | 70.45 |
| Llama-2-7b-hf (Touvron et al., 2023b) | 7.00E+09 | 2.00E+12 | 1.03E+23 | 52.65 | 74.54 | 77.71 | 78.98 | 79.11 | 46.11 | 74.19 | 69.04 |
| Llama-2-13b-hf | 1.30E+10 | 2.00E+12 | 1.88E+23 | 59.47 | 77.53 | 80.58 | 84.23 | 80.52 | 47.34 | 76.16 | 71.98 |
| Llama-3-8B (Dubey et al., 2024) | 8.00E+09 | 1.50E+13 | 9.46E+23 | 58.19 | 77.61 | 80.95 | 82.10 | 80.69 | 47.08 | 77.51 | 72.02 |
| Llama-3.1-8B | 8.00E+09 | 1.50E+13 | 3.85E+24 | 57.85 | 81.19 | 82.05 | 81.91 | 81.01 | 46.98 | 77.19 | 72.60 |
| Llama-3.2-1B | 1.23E+09 | 9.00E+12 | 5.29E+23 | 39.59 | 60.61 | 63.91 | 65.51 | 74.27 | 42.99 | 62.27 | 58.45 |
| Llama-3.2-3B | 3.21E+09 | 9.00E+12 | 1.40E+24 | 50.68 | 71.84 | 72.75 | 76.42 | 77.37 | 47.39 | 71.82 | 66.90 |
| OLMo-1B-hf (Muennighoff et al., 2024) | 1.00E+09 | 2.00E+12 | 1.56E+22 | 34.47 | 57.28 | 61.74 | 63.81 | 75.14 | 42.12 | 60.46 | 56.43 |
| OLMo-7B-hf | 7.00E+09 | 2.46E+12 | 1.26E+23 | 45.14 | 68.77 | 72.45 | 77.13 | 79.43 | 44.52 | 70.96 | 65.49 |
| OLMo-2-0425-1B (OLMo et al., 2024) | 1.00E+09 | 4.00E+12 | 3.04E+22 | 45.39 | 73.36 | 63.03 | 68.71 | 75.63 | 43.76 | 65.90 | 62.25 |
| OLMo-2-1124-7B | 7.00E+09 | 4.00E+12 | 2.03E+23 | 64.51 | 82.87 | 80.00 | 81.93 | 81.01 | 51.33 | 77.03 | 74.10 |
| OLMo-2-1124-13B | 1.30E+10 | 5.00E+12 | 4.67E+23 | 66.13 | 81.31 | 73.91 | 84.99 | 82.15 | 52.05 | 83.03 | 74.80 |
| OLMo-2-0325-32B | 3.20E+10 | 6.00E+12 | 1.30E+24 | 69.45 | 85.94 | 82.81 | 87.33 | 82.97 | 54.25 | 83.90 | 78.09 |
| Qwen3-0.6B-Base (Yang et al., 2025) | 6.00E+08 | 3.60E+13 | 5.31E+23 | 44.80 | 58.00 | 66.82 | 53.46 | 69.80 | 43.30 | 60.46 | 57.09 |
| Qwen3-1.7B-Base | 1.70E+09 | 3.60E+13 | 1.18E+24 | 55.20 | 68.60 | 79.24 | 67.19 | 75.52 | 48.62 | 65.27 | 65.66 |
| Qwen3-4B-Base | 4.00E+09 | 3.60E+13 | 2.19E+24 | 64.42 | 75.93 | 82.91 | 75.64 | 77.86 | 50.00 | 72.61 | 71.34 |
| Qwen3-8B-Base | 8.00E+09 | 3.60E+13 | 3.90E+24 | 67.24 | 79.88 | 83.09 | 79.54 | 79.54 | 54.76 | 77.19 | 74.46 |
| Qwen3-14B-Base | 1.40E+10 | 3.60E+13 | 6.09E+24 | 69.97 | 81.86 | 86.76 | 82.69 | 82.10 | 55.89 | 79.48 | 76.96 |
| Qwen2.5-0.5B (Yang et al., 2024) | 5.00E+08 | 1.80E+13 | 2.04E+23 | 35.24 | 58.54 | 61.47 | 51.83 | 69.80 | 44.17 | 56.59 | 53.95 |
| Qwen2.5-1.5B | 1.50E+09 | 1.80E+13 | 1.38E+24 | 54.86 | 72.10 | 72.48 | 67.86 | 75.90 | 49.08 | 65.27 | 65.36 |
| Qwen2.5-3B | 3.00E+09 | 1.80E+13 | 8.51E+23 | 56.31 | 73.02 | 77.43 | 74.54 | 78.67 | 49.80 | 71.67 | 68.78 |
| Qwen2.5-7B | 7.00E+09 | 1.80E+13 | 3.62E+24 | 63.65 | 77.48 | 84.65 | 80.19 | 79.82 | 54.61 | 76.40 | 73.83 |
| Qwen2.5-14B | 1.40E+10 | 1.80E+13 | 8.58E+24 | 67.58 | 79.25 | 85.35 | 84.21 | 82.43 | 55.48 | 81.06 | 76.48 |
| Qwen2.5-32B | 3.20E+10 | 1.80E+13 | 1.29E+25 | 70.65 | 77.99 | 87.49 | 85.16 | 82.43 | 56.29 | 82.08 | 77.44 |
| SmolLM-135M (Allal et al., 2024) | 1.35E+08 | 6.00E+11 | 1.51E+21 | 32.00 | 56.14 | 60.09 | 42.92 | 68.01 | 39.56 | 52.25 | 50.14 |
| SmolLM-360M | 3.60E+08 | 6.00E+11 | 3.16E+21 | 38.65 | 63.59 | 55.05 | 54.24 | 71.44 | 40.99 | 57.14 | 54.44 |
| SmolLM-1.7B | 1.70E+09 | 1.00E+12 | 1.54E+22 | 49.40 | 73.57 | 66.15 | 67.33 | 75.95 | 43.35 | 61.72 | 62.50 |
| SmolLM2-135M (Allal et al., 2025) | 1.35E+08 | 2.00E+12 | 2.48E+21 | 33.02 | 58.38 | 60.06 | 43.64 | 68.12 | 39.25 | 53.12 | 50.80 |
| SmolLM2-360M | 3.60E+08 | 4.00E+12 | 1.20E+22 | 40.78 | 68.22 | 61.56 | 57.46 | 71.76 | 40.89 | 58.41 | 57.01 |
| SmolLM2-1.7B | 1.70E+09 | 1.10E+13 | 1.21E+23 | 53.50 | 73.27 | 72.32 | 73.16 | 77.53 | 44.52 | 68.35 | 66.09 |
| SmolLM3-3B-Base | 3.00E+09 | 1.12E+13 | 8.56E+23 | 59.81 | 76.85 | 80.49 | 77.18 | 79.11 | 46.78 | 73.40 | 70.52 |
| Zamba2-1.2B (Glorioso et al., 2024) | 1.20E+09 | 3.00E+12 | 3.86E+23 | 53.92 | 66.71 | 70.18 | 72.21 | 77.20 | 46.42 | 68.98 | 65.09 |
| Zamba2-2.7B | 2.70E+09 | 3.00E+12 | 4.77E+23 | 60.67 | 73.82 | 78.07 | 77.72 | 79.49 | 45.50 | 76.01 | 70.18 |
| Zamba2-7B | 7.40E+09 | 2.00E+12 | 7.68E+23 | 68.34 | 80.39 | 83.70 | 83.53 | 80.69 | 49.90 | 79.72 | 75.18 |
| Celerity-300M | 2.71E+08 | 6.34E+10 | 1.47E+20 | 27.82 | 50.63 | 52.75 | 37.57 | 66.21 | 37.77 | 52.25 | 46.43 |
| Celerity-500M | 5.03E+08 | 1.18E+11 | 5.15E+20 | 34.39 | 56.06 | 61.22 | 45.96 | 69.31 | 40.23 | 52.64 | 51.40 |
| Celerity-900M | 9.06E+08 | 2.12E+11 | 1.68E+21 | 39.68 | 64.52 | 47.92 | 55.02 | 72.03 | 41.97 | 58.48 | 54.23 |
| Celerity-1.8B | 1.81E+09 | 4.24E+11 | 6.54E+21 | 48.55 | 70.29 | 65.17 | 64.34 | 75.46 | 42.99 | 60.22 | 61.00 |
| Celerity-3.9B | 3.88E+09 | 9.08E+11 | 2.89E+22 | 54.01 | 75.55 | 66.61 | 72.19 | 77.97 | 44.73 | 65.90 | 65.28 |

- Use a functional form that accounts for training fraction and LR schedule
- Make the *parameters* of this functional form depend on TPP and $\tau$

This led to Eq. (4). Our initial aim here is not to develop the best possible TLC predictor, but to obtain a simple, effective, and interpretable method for extrapolating TLCs, allowing us to test the value of this extrapolation for early stopping in hyperparameter tuning.

We conducted a variety of preliminary experiments at 111M-scale, using the same data as in Sec. 3 (with details in Appendix B.1). As an input to Eq. (4), the LR schedule is normalized to be at 1.0 at its peak. It's also interpreted over training fraction, so from 0.0 to 1.0. Experiments in this data only use fits for linear decay-to-zero schedules. To get an initial sense of how the parameters in Eq. (4) vary, we did a multi-dimensional grid search to determine optimal parameters for each individual

curve, measuring total macroaveraged MAE loss over all 111M-scale TLCs. Over the course of these experiments, we found total MAE did not change substantially when we fixed $m = 0.05$, and we subsequently tuned $\epsilon_1$ and $\epsilon_2$ to small constants in order to avoid boundary effects at $\hat{t} = 0$ and $\hat{t} = 1$ (when $\eta(\hat{t})$ goes to zero). Prior to fitting, training curves were smoothed using a moving average filter covering 12288 sequences (equal to the largest batch size in the dataset), and we ignore error on the first 20% of each curve (around LR warmup when curves are noisy).

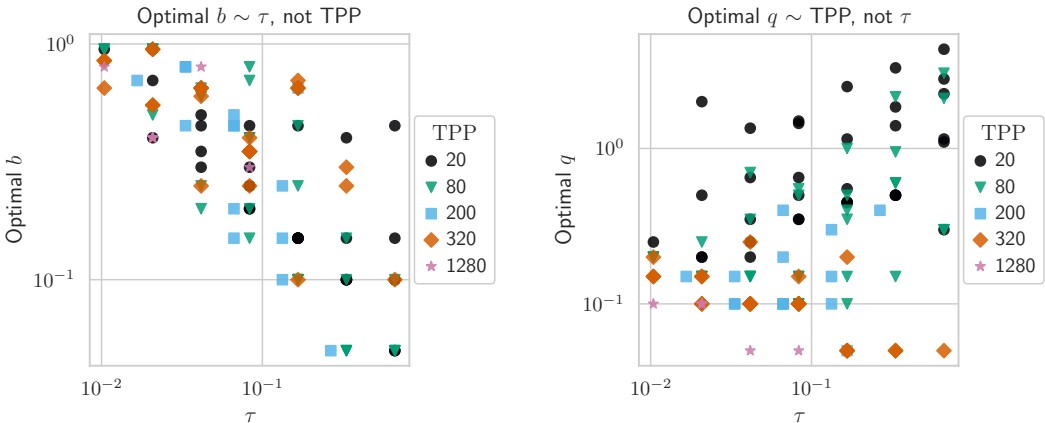

Figure 22: **Trends in fits for training curve prediction.** Optimal per-curve fits (from per-curve grid searches) for Eq. (4): $\hat{\ell}(\hat{t}) \approx 1/m^{0.05} + b \cdot \eta(\hat{t})^q$: $b$ and $q$ parameters. *Left*: Optimal $b$ varies strongly in $\tau$ (Pearson's $r$ = -0.59), weakly in TPP ($r$ = 0.17). *Right*: Optimal $q$ varies somewhat in TPP ($r$ = -0.30), while overall stronger in $\tau$ ($r$ = 0.55), but $\tau$ trends reverse at higher TPP.

Fig. 22 shows the optimal fits for $b$ and $q$ when each curve is fit independently. For **optimal** $b$, we found that correlation in $\tau$ was much stronger than correlation in TPP (Pearson's $r$ = -0.59 for $\tau$, $r$ = 0.17 for TPP). On the other hand, while **optimal** $q$ seems to increase with $\tau$ for TPP = 20, the relationship with $\tau$ at other TPP appears random. Furthermore, note larger TPP values do correspond to lower optimal $q$ ($r$ = -0.30). Based on these fits, we hypothesize we could obtain reasonable predictions by fitting $b$ as a power law in $\tau$, and $q$ as a power law in TPP:

$$b = b_{\text{const}} \cdot \tau^{b_{\text{exp}}}, \quad q = q_{\text{const}} \cdot \text{TPP}^{q_{\text{exp}}} \tag{24}$$

As noted in Sec. 5. Also, as reported in that section, we developed an alternating greedy optimization procedure to fit these four parameters, exponentially reducing the cost of the grid search space.

**Results.** We first note that the fits improve over the iterations of our alternating grid search procedure, demonstrating that optimal parameters of the power laws do depend on each other, and can reach stable fits through iterative alternating fitting.

Table 11: **Predictions improve with scale**: fit at 111M scale, evaluated at larger scales.

| Evaluation scale | MAE | Number of evaluation curves |
|---|---|---|
| 111M* (fitting points) | 1.37% | 112 |
| 266M | 0.75% | 40 |
| 610M | 1.07% | 102 |
| 1.7B | 0.66% | 21 |
| 3.3B | 0.54% | 7 |

Table 11 and Table 12 are the tables discussed in the main paper, showing how fits obtained at 111M perform at other scales (Table 11), and how different fitting procedures perform on the 610M-scale evaluation data (Table 12). Fitting $b$ and $q$ with the optimum values *per-curve* (i.e., *oracle* fits) achieves an MAE of 0.504%, roughly half that of the dual power law extrapolations.

Table 12: **Separate power laws for $b$ and $q$ work well**: fit at 111M scale (112 TLCs), evaluation at 610M (102 TLCs).

| Method for estimating $b$ | Method for estimating $q$ | MAE |
|---|---|---|
| Global fixed optimum | Global fixed optimum | 3.03% |
| Global fixed optimum | $q = \text{PowerLaw}(\text{TPP})$ | 3.35% |
| $b = \text{PowerLaw}(\tau)$ | Global fixed optimum | 2.08% |
| $b = \text{PowerLaw}(\tau)$ | $q = \text{PowerLaw}(\text{TPP})$ | 1.07% |
| $b = \text{PowerLaw}(\tau, \text{TPP})$ | $q = \text{PowerLaw}(\tau, \text{TPP})$ | 1.07% |

### D.2 Early stopping in tuning: further results

In this section we describe some further early stopping experiments, and present additional evaluation metrics.

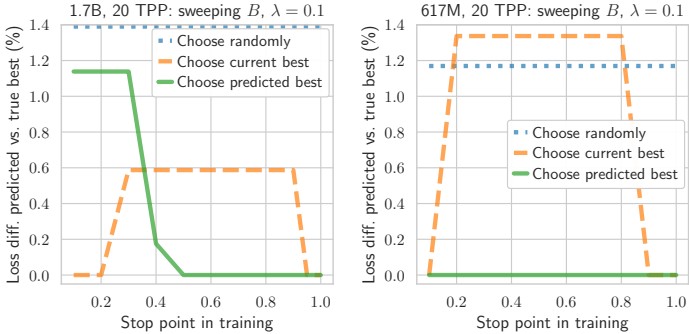

Figure 23: **Early stopping comparison: further setups.** Companion to Fig. 9, now comparing early stopping accuracy (final loss of predicted vs. actual best) for $B$ sweeps at 1.7B (*left*) and 617M (*right*) (both 20 TPP). *Current best* works well very early, but is worse for most of training.

Fig. 23 evaluates early stopping strategies in batch-size sweeps at a fixed $\lambda$ value. Fig. 23, *left*, uses the same data as in Fig. 7, *left*. While we do not advocate keeping $\lambda$ fixed during $B$ sweeps in practice, this data can nevertheless serve to evaluate prediction of early winners in tuning. Both of these plots exhibit the phenomenon also observed in Fig. 9, *right*: choosing the current best setting after LR warmup, as was done in Falcon (Almazrouei et al., 2023), is better than selecting the best during the middle of training. However, as seen in Fig. 9, *left*, this method is not always successful. In Fig. 23, *left*, choosing the extrapolated best setting outperforms choosing the current best from 40% of training, while it picks the correct winner from the beginning in Fig. 23, *right*.

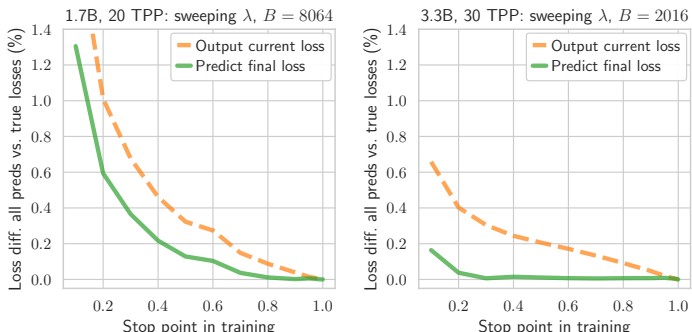

Figure 24: **Early stopping comparison: MAE at 1.7B, 3.3B: $\lambda$ sweeps.** Mean absolute error of all predicted final losses, comparing taking current loss vs. extrapolating final loss.

In many cases, rather than caring purely about which setting is best, we care about the actual projected final loss. This may be useful for fitting scaling laws, or for helping practitioners reason about

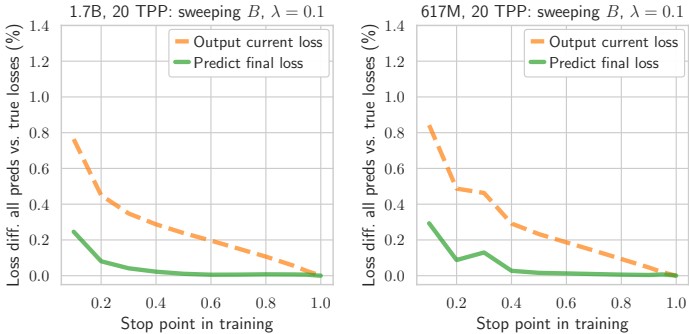

Figure 25: **Early stopping comparison: MAE at 1.7B, 617M: $B$ sweeps.** Mean absolute error of all predicted final losses, comparing taking current loss vs. extrapolating final loss.

the trade-offs of, for example, greater throughput from larger $B$ vs. suffering higher final loss. We therefore evaluated the same four hyperparameter sweeps above, but now evaluating the average loss difference between the predicted final loss and the true final loss for all curves. The baseline chooses the current loss for each curve at the given training fraction, which will overestimate the final loss. Results in Figs. 24 and 25 show that in three of four cases, extrapolating the final loss using our predictive form results in *much* smaller average error than using the current value.

The only instance where predicting the final loss incurred significant error was the 1.7B, 20 TPP model with $B = 8064$. We note that the TLCs are very noisy at this high batch size across almost all $\lambda$ settings and therefore it is evidently challenging to align the in-progress training runs to the predicted TLC. Increasing smoothing reduces the predicted error somewhat, but the primary issue is that the noise affects the TLC mainly in the first 60% of training, thus distorting even the smoothed loss from the universal trajectory. Accurate prediction in the presence of loss spikes is an acknowledged limitation of our methodology (Appendix A).

## D.3 OPTIMAL AND SUBOPTIMAL TLCS AS TPP SCALES

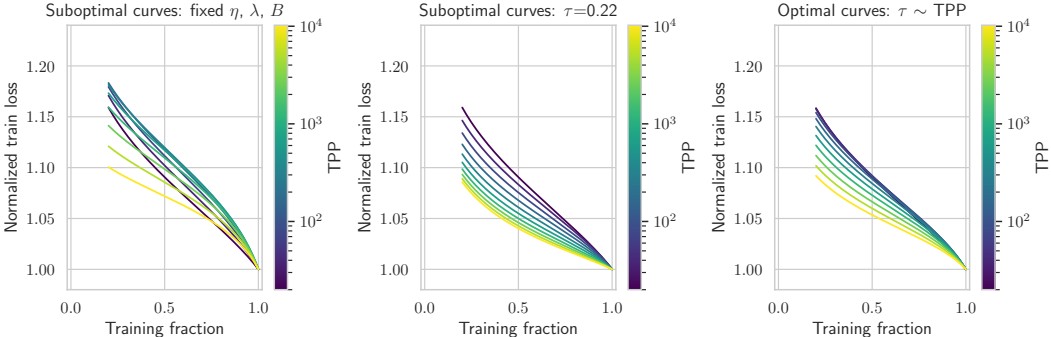

Figure 26: **Evolution of train curve shape.** *Left*: When TPP is scaled but $\eta$, $\lambda$ and $B$ are held constant, curve shape varies significantly. *Middle*: When $\tau$ is instead held constant, shape evolves more gradually. *Right*: When $\tau$ scales with TPP according to established power laws, curves maintain their concave structure.

Given a fitted predictive form (Eq. (4)), it is natural to ask how TLC shape varies as TPP increases, under various hyperparameter (HP) scaling strategies. In this section, we consider three scenarios:

1. No adjustment to any HPs: basically standard practice under $\mu$P until very recently.

2. Maintain constant $\tau$: i.e., following the prescription of Wang & Aitchison (2024).

3. Optimize $\tau$: adjust $\tau$ for each TPP setting following the $\tau$ power law of Bergsma et al. (2025a).

Results in Fig. 26 demonstrate that, with no HP adjustments, curve shape changes substantially across TPP (*left*). Fixing $\tau$ results in more consistent shapes (*middle*), but only when $\tau$ is scaled for TPP do curves maintain their characteristic concave shape, with a noticeable drop near the end of training (*right*). One may view this final period as the *annealing* phase of training, or the phase where variance is reduced and we descend the valley into the river (Wen et al., 2024). As TPP increases, we must reduce $\tau$ correspondingly to prioritize exploration for the majority of training, enabling this final descent only in the final phases.

