# OpenReview forum: "Scaling with Collapse: Efficient and Predictable Training of LLM Families"
_ICLR.cc/2026/Conference — ICLR 2026 Poster_

### Official Review · Reviewer_SS9H · 2025-10-17

**Soundness:** 4
**Presentation:** 4
**Contribution:** 3
**Rating:** 8
**Confidence:** 2

**Summary:**

The paper studies how the training of large language models can be made more predictable and efficient. The authors observe that when optimization hyperparameters are set correctly, the normalized training loss curves for models of different sizes collapse onto a single curve. This loss collapse suggests a consistent and predictable scaling behavior across model sizes.

The paper uses this finding to propose practical tools for large-scale training, e.g., using loss collapse for early diagnostics and for tuning hyperparameters more efficiently. The authors also introduce a new model family, Celerity, to demonstrate that these principles can lead to faster and more compute-efficient training.

The results are supported by experiments across multiple model sizes, and the figures clearly show the observed alignment of loss curves.

**Strengths:**

(I am not very familiar with this specific literature. My assessment is therefore based on the content and presentation of this paper itself, rather than detailed knowledge of prior work. Judging solely by the paper, it appears to be a solid piece of work.)

Strengths:

- The paper addresses an important and very active research area (making large-scale model training more predictable and compute-efficient through analysis of loss-curve regularities).

- The authors present convincing experimental evidence that normalized training loss curves collapse across model scales when key hyperparameters are properly aligned. The results span a range of model sizes.

- The idea of using loss-curve collapse as a diagnostic tool and a guide for efficient hyperparameter tuning is valuable in large-scale training settings.

- The paper is generally well-written and easy to follow. The figures are informative and communicate the main ideas well.

- The introduced Celerity model family shows that the proposed ideas can work well in practice.

**Weaknesses:**

- The paper's main ideas are based mostly on experiments, and the theoretical explanations are limited.

- The experiments focus on GPT-like architectures trained with AdamW. It is not clear whether the same collapse behavior holds for other architectures or optimizers.

**Questions:**

- Have you tested the method with more recent state-of-the-art optimizers, e.g. Muon [1]?

- Does the same loss collapse happen for validation loss, or only for training loss?

[1] Keller Jordan, Yuchen Jin, Vlado Boza, You Jiacheng, Franz Cesista, Laker Newhouse, and Jeremy Bernstein. Muon: An optimizer for hidden layers in neural networks, 2024.

---

> ### Author Response · Authors · 2025-11-20
> **Response to Reviewer SS9H (1/1)**
>
> Thank you very much for your helpful and encouraging review.  We are very glad you found the experimental evidence, presentation, and overall significance to be strong.
>
> **Regarding testing with other optimizers**: We have not yet tested collapse with optimizers such as Muon, but this is an important future direction.  As noted in Appendix A (lines 870-873), we acknowledge our experimental focus on AdamW as a limitation. That said, Muon variants typically retain weight decay for LLM training (e.g., MuonClip in Kimi K2), and these update rules can be expressed in EMA form, analogous to AdamW. We therefore expect the EMA timescale τ to remain a primary control of TLC shape, and thereby expect collapse to extend to these optimizers. Verifying this empirically is an important next step, and we appreciate you highlighting it.
>
> **Does collapse hold for validation loss?**: Yes. Although we only evaluated validation loss at a subset of training steps, we *did* observe collapse in validation curves across model sizes. This is consistent with the typical behavior in LLM pre-training, where training loss tracks validation performance under stationary data distributions. We will include these validation-collapse results in the final paper for clarity and completeness. We will also note in the paper that validation collapse could be a particularly valuable diagnostic if late-stage annealing or curriculum strategies distort training curves. Thank you for raising this point!

---

### Official Review · Reviewer_J2cj · 2025-10-27

**Soundness:** 4
**Presentation:** 3
**Contribution:** 4
**Rating:** 8
**Confidence:** 5

**Summary:**

The authors use recent work on learning curve universality in compute-optimally trained networks, combined with other observations on the timescales of weight decay to come up with a training/scaling procedure for language models that takes advantage of these effects. They show that correct selection of the weight decay timescale, via either the (coupled) weight decay parameter, learning rate, or batch size/total number of tokens can reshape the loss curve in a consistent manner. They also show that the collapse principle can be used to predict early training failures, and to develop a model that can use detailed data from small scale training runs to predict the results of large scale training runs by 10-30% of the length of the larger scale training run.

**Strengths:**

This paper well-integrates key recent results on scaling behavior and uses it to provide a compelling angle on how to scale large models. The experiments are generally thorough and the paper is overall well-presented. The authors provide strong evidence that the universality of learning curves, combined with properly chosen scaling procedures, can be used to train compute-optimally with less risk than with previous methods.

**Weaknesses:**

One thing that is not immediately clear to me is the utility of the model presented in section 5. It seems like the best procedure would be:

* Search over hyperparameters at small scale (e.g. 100M).
* Match parameters at larger scale.
* Carry out partial training runs at large scale.
* Use the partial training runs to select the best run and finish it.

It seems to me that even with the model, one may have to run many training runs at the largest scale in order to obtain the normalization factor, aka the finalized loss.

**Questions:**

Can you explain the details of the point raised in weaknesses? This is key to the underpinning of the model/fitting procedure proposed in Section 5; right now it is still unclear to me how the proposed model allows for optimization of the unnormalized loss at larger scale.

With regards to the learning curve predictions: can this model be used to predict the results of varied learning rate schedules, or is it mainly suited to predicting the correct overall timescales?

---

> ### Author Response · Authors · 2025-11-20
> **Response to Reviewer J2cj (1/1)**
>
> Thanks so much for your helpful review, and your kind words regarding the paper's thorough experiments, strong evidence, and overall compelling story.
>
> **Regarding the predictive model in Section 5**: Thanks for raising this, it was definitely not explained sufficiently in the submission. We briefly noted that small-scale runs could be used to obtain the normalized universal curves (lines 385-386), but the section would be clearer if we explicitly articulated the model-free procedure (which you summarized perfectly). We agree it would be an effective and intuitive way to tune. Using the predictive model is indeed unnecessary if we have already trained, or are willing to train, the small-models with corresponding τ and TPP. We propose to revise this section as follows:
>
> > **Exploiting collapse for early stopping.**
> >
> > Suppose we wish to find the optimal setting of a hyperparameter in large-scale training. Naively, we could sweep across settings of the hyperparameter and train a large-scale model to completion at each setting; the lowest final loss would identify the best choice. Alternatively, rather than training to completion, we propose a procedure to *infer* the final loss values from partial training runs.  We do this by exploiting the collapse phenomenon as follows:
> >
> > 1. For each large-scale setting in the sweep, first identify the corresponding **TLC controls**, i.e, the LR schedule, τ, and TPP.
> > 2. Train a smaller (e.g., 100M-parameter) model for each unique combination of TLC controls.
> > 3. Normalize these small-run loss curves to obtain the **normalized universal TLCs** corresponding to each set of controls.
> > 4. Perform **partial** training runs for the large-scale sweep (e.g., 30% of tokens) at each hyperparameter setting.
> > 5. For each partial large-scale loss curve, use the corresponding universal TLC to *predict* the final loss value.  Do this by determining the divisor **L(T)** that maximally *aligns* the partial curve with the same segment of the corresponding universal TLC. These divisors thus act both as normalizing constants *and*, by construction, as calibrated **extrapolations** of the final loss.
> > 6. Select the optimal hyperparameter setting using the lowest predicted final loss.
> >
> > **Predicting the normalized universal TLCs.**
> >
> > In some cases, we may already have many small-scale runs, but not loss curves for each of the *specific* TLC controls corresponding to our large-scale sweep. In this situation, we hypothesize that a **parametric surrogate model** can generate high-fidelity normalized universal TLCs as a function of τ and TPP.  This allows us to obtain the normalized TLC shapes required for Step 5 of the above procedure, *without* training small-models at exactly-matching controls.  Fitting such a surrogate lets us leverage our broader TLC dataset, and enables estimation of normalized TLC shapes that go beyond trained regimes (see Appendix D.3 for an example).
> >
> > [we next provide the functional form and explain how we fit it]
>
> **Regarding loss prediction across learning rate schedules.** Thank you for this question - it raises an important point. Our functional form *does* incorporate the learning rate schedule η(t), so in theory we can fit and predict the normalized universal TLCs for any schedule. However, because these predictions give only *normalized* loss, they do not by themselves identify the *optimal* schedule.  To choose among schedules, we *could* use them—or training runs with each schedule at small scale—as part of the alignment+extrapolation procedure that we enumerated above.
>
> Interestingly, recent work has shown that the optimal LR decay ratio is TPP-dependent (https://arxiv.org/abs/2502.15938). This suggests an intriguing question for future work: can LR decay functions be tuned at *small scale* (at different TPP values), and then transferred directly to larger models?  We will list this specifically as promising future work.  Thanks again for stimulating this discussion.

---

> > ### Comment · Reviewer_J2cj · 2025-11-20
> > **Response to authors**
> >
> > I thank the authors for their response! I really like the proposal for the edit, I think this will greatly improve the paper. Thanks also for the clarifications on my questions.

---

### Official Review · Reviewer_ak8p · 2025-10-30

**Soundness:** 4
**Presentation:** 2
**Contribution:** 3
**Rating:** 8
**Confidence:** 4

**Summary:**

This paper demonstrates that training loss curves (what they call TLCs) of LLM training collapse onto a universal trajectory when normalized, provided that three key factors match: the AdamW timescale τ, the tokens-per-parameter ("TPP") ratio, and the learning rate schedule. The authors show that this collapse phenomenon emerges naturally when optimization hyperparameters are set optimally for a given data budget. They then train Celerity, a family of LLMs trained at fixed TPP with optimal τ, demonstrating the collapse across scales, collapse residuals as early diagnostics for training issues, and a predictive model enabling early stopping in hyperparameter tuning.

**Strengths:**

Quality: The biggest strength of the paper is that the experimental work is thorough and well-executed. Every major ablation you would need is provided, with lots of comparisons (e.g., varying batchsizes, LR sweeps, weight decay, ...). The appendix provides even more.

Originality: The work follows a continuous line of research about stable and predictable dynamics, and specifically extends the findings of Qiu et al. to LLM training. As such, the contribution is clear: to me, it provides sound evidence of a phenomena we previously believed to be true. The application to early stopping/hyperparameter testing is interesting and well motivated, and the case study with the 1.8B run recovery is cool.

Significance: Predictability of large scale LLM training runs is extremely important, e.g., being able to reason about optimality, avoid wasting computational resources, or restart

**Weaknesses:**

The authors are already well aware of the main limitations: the results are done with a fixed (but certainly standard) way of pretraining, i.e., single epoch, AdamW, moderate dataset sizes ("TPP"). The conclusions might not hold or look different for setups with much longer training, data curriculums or optimizers and schedules. Especially longer training (higher "TPP") would be relevant (is there a point where i breaks down?) and extrapolations beyond just training loss. That being said, the paper is already extensive enough and cannot cover too many topics at once.

Minor: Celerity uses a much more refined datamix than just Slimpajama, which makes some comparison (e.g. to BTLM) somewhat hard. But that is discussed (and represents a general improvement of pretraining strategies including data+optimization).

Besides, I believe the biggest weaknesses revolve around accessibility, which I would cluster into two points: readability and availability of resources. First, as mentioned above, the paper is the newest in a serious of publications around LLM training predictability and scaling, which all build upon each other. I understand the authors have an in-depth understanding of this area, but the paper is extremely dense, with lots of notation and terms that make it daunting and complex to grasp for new readers. I think this presentation can be improved. Second, and very related, the authors have not published the code or any of the models as far as I can tell. Since the exact setup is quite unique (e.g., CompleteP), releasing those artifacts would be crucial in enabling better study of the entirety of methods that are included in this line of work.

**Questions:**

* Maybe I am missing something, or why are the Llama3 and Qwen3 models not included in any figure comparison? The models are in the table, yet their points miss in plots.

* I would highly encourage the authors to include the list of benchmarks in the Fig. 2 caption, link to them, or make it clearer what is being measured. It is a good overview but I would initially suspicious because I could not immediately find what accuracy was actually measured.

* More broadly, figure captions are not self-contained, which makes it hard to grasp details or what exact setup was used; instead I often needed to jump back and forth between text and figures.

* The bias-variance tradeoff is somewhat glossed over, and without spending more extensive time studying the paper, I could not grasp this discussion exactly. Could you potentially clarify directly?

* Figure 15 shows CompleteP outperform μP quite substantially in terms of not only offset but slope. You provide some explanations, but have you verified experimentally how the μP transfer is suboptimal in your LR setup?

---

> ### Author Response · Authors · 2025-11-20
> **Response to Reviewer ak8p (1/2)**
>
> Thank you very much for your detailed and constructive feedback, and for generously noting the paper's experimental rigor, comprehensive ablations, and important findings.  We address your readability/accessibility concerns and specific questions below.
>
> ### Readability
>
> We agree that the current draft is dense. We will take the following steps to make the exposition more accessible:
> - **Add an intuitive overview** (Sec. 2): We will include a brief, non-technical preamble summarizing loss curve collapse and the roles of TPP, τ, and the learning rate schedule, before introducing notation.
> - **Add a table of symbols**: We will add a concise table listing the main quantities used throughout the paper (e.g., L(t), ℓ(t), TPP, τ), to reduce back-and-forth navigation and improve clarity.
> - **Expand the "Key takeaways"**: We will expand the Key Takeaway boxes in Secs. 3-5 to restate the central points in plainer language and make the flow of ideas clearer.
> - **Clarify the bias-variance interpretation of τ**: we will amend the section "Explaining effect of τ" to begin:
>
> > **Using bias and variance to explain the effect of τ.**
> >
> > Classical analyses of stochastic optimization (e.g., Bottou et al., 2018, D'Angelo et al., 2023) model loss as an evolving sum of two components:
> > - a **bias** term, reflecting dependence on the initial weights, and
> > - a **variance** term, reflecting noise from stochastic gradients.
> >
> > Early in training the bias term is typically much larger; later, the variance term dominates the remaining loss. This analysis helps motivate the learning-rate schedules used in practice, where large early LRs reduce bias and smaller later LRs suppress variance.
> >
> > In our setting, the AdamW timescale τ controls bias vs. variance reduction through its impact on the exponential-moving-average over weight updates (Sec. 2). A smaller τ gives the EMA a **short memory**, so updates depend mainly on recent gradients. This reduces bias (and hence loss) quickly but leads to a higher variance floor. A larger τ gives the EMA a **longer memory**, averaging over many past gradients and thereby reducing variance more effectively, though with slower early progress in bias reduction (slower loss improvement). Experimentally, the training loss curves reflect this τ-mediated pace of early vs. later loss reduction (e.g., see the left panel of appendix Fig. 10), dropping faster when τ is small.
> >
> > This intuition is formalized in Appendix B.3 via a simple noisy-quadratic model, which decomposes the expected loss into an exponentially decaying bias term and a variance term that rises to a floor proportional to 1/τ. [The remainder of this subsection gives the full expression, explains how LR schedules create a *dynamic* τ, and how this affects loss curves, etc.]
>
> - **Improve figure captions**: We will make figure captions more self-contained, especially for Fig. 2 and the collapse figures, and we will list the benchmark suite used in Fig. 2 directly in the caption to avoid requiring cross-referencing.
> - **Section 5 readability**: We will also apply the clarifications described in our response to Reviewer J2cj regarding the presentation of Sec. 5.
>
> The extra page in the camera-ready allows us to incorporate these improvements without removing technical content.
>
> ### Accessibility
>
> While we have provided comprehensive information on the data-mix, hyperparameters, architecture and other pre-training details needed for replication, we do agree that sharing other resources would help maximize the impact of our work.  To this end:
> - In the camera-ready version, we will link to an implementation of CompleteP so that others can review the code changes that we used for Celerity training.
> - We are also seeking approval to release models trained with the Celerity recipe, as well as other artifacts like loss curve data from our ablations.

---

> > ### Author Response · Authors · 2025-11-20
> > **Response to Reviewer ak8p (2/2)**
> >
> > ### Additional questions
> >
> > - **Llama-3 and Qwen-3 not plotted in Fig. 2**: Thank you for catching this! We will revise the paper to clarify that we only *plotted* results for models with sufficiently precise and complete training details. E.g., Llama-3.2 reports using "Up to" 9T tokens (https://console.cloud.google.com/vertex-ai/publishers/meta/model-garden/llama3-2) while Llama-3.1 and Qwen-3 sizes are reported "approximately." Moreover, it is unclear from the papers whether the full pre-training datasets (i.e., used to train the *flagship* models) were also used for the smaller models. We will separately identify the results in Table 9 that are based on assumptions, i.e., the assumption that the models *do* use the full (approximately-reported) datasets. Note that if these assumptions hold, the smaller Llama-3 and Qwen-3 models would be rather compute-*inefficient*, e.g., they would all be beyond the plotting range of Fig. 2 (i.e., >10^24 FLOPs) and score well below the Celerity extrapolation (and Gemma results) in Figure 16.
> > - **CompleteP vs. μP (Fig. 15)**: This is also a very good point. Currently, the figure caption implies that differences are fully due to poor HP transfer, but we will clarify that we have *not* run the sweeps to verify this experimentally. In fact, we believe differences are due both to better HP transfer *and* better efficiency of CompleteP models. Prior studies have shown poor scaling law fits when scaling in the standard parameterization vs. μP, and we believe a similar phenomenon is now happening across depth. Meanwhile, results in Dey et al. 2025 (Fig. 4) suggest that CompleteP models exhibit better scaling behavior than μP, even when both are more comprehensively tuned. We will revise the figure caption and the discussion to expand on these points.
> >
> > We really appreciate the reviewer’s suggestions; improving the clarity of the exposition is a high priority for us and we believe addressing these comments will strengthen the paper significantly.

---

### Author Response · Authors · 2025-12-02
**Author Final Remarks**

Thank you again to the reviewers for their thoughtful and constructive feedback, and to the (new) AC for taking on this additional responsibility in light of the recent OpenReview incident.

Our paper shows that training loss curves *collapse* across LLM scales when training at a fixed tokens-per-parameter ratio and choosing the AdamW timescale τ optimally for that ratio. This collapse acts as both a marker of compute-efficient training and as a practical tool for diagnostics, tuning, and early stopping. All three reviewers found the experimental evidence, significance, and overall contribution to be strong.

We have proposed several concrete changes to address reviewer concerns regarding presentation. These include:
- adding an intuitive overview in Section 2 and a table of symbols,
- expanding the Key Takeaways and improving figure captions (including listing benchmarks in Fig. 2),
- rewriting key explanatory passages (i.e., "Using bias and variance to explain the effect of τ." and "Exploiting collapse for early stopping." in the rebuttals below),
- providing additional results (e.g., the observed validation-loss collapse) and clarifying certain limitations and comparisons (e.g., CompleteP vs. μP).

We also plan to link to additional resources in the camera-ready version, including an implementation of CompleteP, and we will seek approval for releasing model checkpoints and loss-curve data.

We believe these clarifications and additions will further strengthen the paper for the final version, and are grateful for everyone's participation in this process.

---

### Meta-Review · Area_Chair_JvDA · 2026-01-06

**Summary:**

**There is strong consensus across all reviewers that this paper makes a significant, timely, and well-executed contribution to effective LLM training.** In particular, reviewers found the empirical evidence for loss-curve collapse to be convincing and carefully validated across scales. They also recognized collapse as a useful diagnostic and predictive tool for detecting training pathologies, guiding hyperparameter tuning, and enabling early stopping. In addition, the introduced Celerity model family demonstrates that the proposed ideas are effective in practice.

All reviewers rated the paper highly (scores of 8). Moreover, the main concerns raised were well addressed in the rebuttal.

**Reviewer Concerns:**

**Reviewer ak8p’s concerns:**

(1) The concerns regarding readability were addressed by the authors through proposed substantial presentation improvements, including the addition of an intuitive overview in Section 2, a table of symbols, expanded “Key Takeaways”, and a rewritten bias–variance tradeoff explanation.

(2) Regarding the availability of resources, the authors committed to releasing the code (CompleteP) and to seeking approval to release models trained with the Celerity recipe, as well as other artifacts such as loss curve data from the ablation studies.

(3) The remaining questions were addressed in the rebuttal, including a clear explanation of why LLaMA-3 and Qwen-3 were excluded from the plots, and a clarified interpretation of CompleteP versus μP in Figure 15.

**Reviewer J2cj’s concerns:**

The concerns raised were addressed in the rebuttal, and Reviewer J2cj expressed satisfaction with the authors’ response.

**Reviewer SS9H’s concerns:**

The questions regarding other optimizers and validation loss were addressed in the rebuttal, while the concerns related to the limitations of the theoretical explanations and the question about other architectures listed under Weaknesses remain outstanding.

**Reviewer Scores:**

**Reviewer J2cj** explicitly expressed satisfaction with the authors’ response and maintained a score of 8.

I believe that **Reviewer ak8p** and **Reviewer SS9H** would not have changed their scores, as their main concerns were addressed in the rebuttal. Both reviewers initially assigned a score of 8 and did not indicate any intention to increase or decrease their evaluations.

---

### Decision · Program_Chairs · 2026-01-26

Accept (Poster)